# Identification of a GABAergic neural circuit governing leptin signaling deficiency-induced obesity

**Yong Han, Yang He, Lauren Harris, Yong Xu, Qi Wu***

USDA/ARS Children's Nutrition Research Center, Department of Pediatrics, Baylor College of Medicine, Houston, United States

**Abstract** The hormone leptin is known to robustly suppress food intake by acting upon the leptin receptor (LepR) signaling system residing within the agouti-related protein (AgRP) neurons of the hypothalamus. However, clinical studies indicate that leptin is undesirable as a therapeutic regiment for obesity, which is at least partly attributed to the poorly understood complex secondary structure and key signaling mechanism of the leptin-responsive neural circuit. Here, we show that the LepR-expressing portal neurons send GABAergic projections to a cohort of $\alpha3$-GABA$_A$ receptor expressing neurons within the dorsomedial hypothalamic nucleus (DMH) for the control of leptin-mediated obesity phenotype. We identified the DMH as a key brain region that contributes to the regulation of leptin-mediated feeding. Acute activation of the GABAergic AgRP-DMH circuit promoted food intake and glucose intolerance, while activation of post-synaptic MC4R neurons in the DMH elicited exactly opposite phenotypes. Rapid deletion of LepR from AgRP neurons caused an obesity phenotype which can be rescued by blockage of GABA$_A$ receptor in the DMH. Consistent with behavioral results, these DMH neurons displayed suppressed neural activities in response to hunger or hyperglycemia. Furthermore, we identified that $\alpha3$-GABA$_A$ receptor signaling within the DMH exerts potent bi-directional regulation of the central effects of leptin on feeding and body weight. Together, our results demonstrate a novel GABAergic neural circuit governing leptin-mediated feeding and energy balance via a unique $\alpha3$-GABA$_A$ signaling within the secondary leptin-responsive neural circuit, constituting a new avenue for therapeutic interventions in the treatment of obesity and associated comorbidities.

*For correspondence:
qiw@bcm.edu

**Competing interest:** The authors declare that no competing interests exist.

## Editor's evaluation

Leptin is a fat-derived hormone that curbs appetite, and mutation of leptin causes obesity and diabetes. This manuscript investigates leptin-responsive neural circuits, revealing a key inhibitory connection from leptin-sensitive neurons in the arcuate nucleus of the hypothalamus (AGRP neurons) to neurons in the dorsomedial hypothalamus. Toggling this inhibitory connection impacted leptin effects on feeding and metabolism.

## Introduction

Diet-induced obesity (DIO) drastically increases susceptibility to the development of metabolic, cardiovascular, and neurological diseases, as well as the recent coronavirus disease (*Kahn et al., 2006*; *Ren et al., 2021*; *Clemmensen et al., 2020*; *Xia et al., 2021*; *Benito-León et al., 2013*). Leptin, a hormone secreted by white adipocytes, interacts with receptors in the brain to communicate peripheral fuel status, suppress appetite following a meal, promote energy expenditure, and maintain blood glucose stability (*Friedman, 2009*; *Friedman, 2014*; *Flak and Myers, 2016*; *Friedman, 2019*; *Knight*

et al., 2010; Berglund et al., 2012; Caron et al., 2018). One of the key pathological characteristics of DIO is impaired metabolic homeostasis (Considine et al., 1995; Maffei et al., 1995). There is much evidence that central leptin signaling plays a vital role in the regulation of body weight, primarily via action within the arcuate nucleus (ARC)(Friedman, 2014; Flak and Myers, 2016; Minokoshi et al., 2002; Huynh et al., 2010; Pereira et al., 2019; Xu et al., 2018; Varela and Horvath, 2012; Vong et al., 2011). The long-form leptin receptors are abundantly expressed in the agouti-related protein (AgRP) neurons of the ARC (Xu et al., 2018; Baskin et al., 1999; Xu and Xie, 2016; Scott et al., 2009; Scott et al., 2011). Previous studies demonstrate that, under the physiological condition, leptin directly inhibits leptin receptor (LepR)-expressing AgRP neurons (Cowley et al., 2001; van den Top et al., 2004). However, clinical studies suggest that treatment of leptin in common obese patients has limited efficacy (Heymsfield et al., 1999; Farr et al., 2015; Hinney et al., 2022). The neural-circuit and transmitter-signaling mechanisms underlying leptin-mediated control of body weight and energy balance are not established.

AgRP neurons play fundamental roles in regulating feeding behavior and body weight by releasing inhibitory neuropeptide Y (NPY), AgRP and GABA transmitters into many downstream brain areas (Kishi and Elmquist, 2005; Han et al., 2021b; Xia et al., 2021). The afferents to these AgRP neurons and their postsynaptic targets have been identified as key players in the regulation of energy balance and systemic insulin sensitivity (Myers et al., 2009; Gropp et al., 2005; Luquet et al., 2005; Aponte et al., 2011; Könner et al., 2007; Steculorum et al., 2016). Leptin acts to decrease food intake and promote energy expenditure by suppressing the activity of AgRP neurons (Cowley et al., 2001; van den Top et al., 2004; Halaas et al., 1997; Friedman and Halaas, 1998). Selective ablation of LepR in AgRP neurons gives rise to an obese phenotype and diabetes (Xu et al., 2018; van de Wall et al., 2008). Although the LepR signaling within AgRP neurons has been implicated as the dominant component for the regulation of metabolic homeostasis, the underlying neural circuit mechanism is still poorly understood. We hypothesize that identification of the critical transmitter signaling components underlying the leptin-responsive neural circuit is crucial for the development of more efficient therapeutics for obesity.

Emerging data suggest a critical role of GABA signaling on the control of feeding behavior and energy homeostasis (Vong et al., 2011; Cone, 2005; Meister, 2007; Morton et al., 2006; Saper et al., 2002; Wu and Palmiter, 2011; Liu et al., 2012; Wu et al., 2013; Wu et al., 2009; Wu et al., 2012; Rossi and Stuber, 2018; Richard, 2015; Meng et al., 2016; Garfield et al., 2016; Pool et al., 2014; Cai et al., 2014; Krashes et al., 2013). Numerous studies suggest that central $GABA_A$- and $GABA_B$-receptor signaling exert prominent influences on feeding behavior under various metabolic states (Berridge, 2009; Cooper, 2005; Duke et al., 2006; Martire et al., 2010; Peciña and Berridge, 1996; Strader et al., 1997; Berner et al., 2009). For example, pharmacological activation of $GABA_A$-receptor signaling in the hindbrain parabrachial nucleus (PBN) enhances the positive hedonic perception of tastes and foods, thereby promoting food intake and motivational response to food reward (Berridge, 2009; Cooper, 2005; Peciña and Berridge, 1996; Berridge and Peciña, 1995; Higgs and Cooper, 1996; Söderpalm and Berridge, 2000; Berridge and Treit, 1986). Utilizing advanced genetic techniques, we have shown that GABAergic signaling from AgRP neurons plays a bidirectional, post-developmental role in the regulation of food intake and body weight (Meng et al., 2016). Our recent work shows that the α5-containing $GABA_A$ receptor signaling within the bed nucleus of the stria terminalis neurons reciprocally regulates mental disorders and obesity, implying that $GABA_A$-receptor signaling exerts a role in controlling obesity and comorbid diseases (Xia et al., 2021).

In this report, we examined the neural-circuit mechanism underlying leptin action upon the AgRP neurons using a newly established and robust method of rapid inactivation of LepR signaling within AgRP neurons (Meng et al., 2016). We found that the LepR in AgRP neurons plays a pivotal role in the control of obesity and glucose homeostasis. Moreover, a subset of leptin-responsive AgRP neurons send GABAergic projections to a group of α3-containing $GABA_A$-expressing neurons in the DMH for regulation of leptin-mediated metabolic homeostasis. These findings suggest that the identification of key $GABA_A$ signaling pathways within the post-synaptic target of a novel leptin-responsive GABAergic neural circuit forebode more efficient treatments for obesity.

# Results

To test the role of LepR in AgRP neurons, we applied our newly developed conditional knockout approach by generating two different lines of mice: $Agrp^{nsCre/+}::Lepr^{lox/lox}::Neo^{R}::Rosa26^{fs-tdTomato}$ mice, termed the knockout group (Agrp-Lepr KO), and $Agrp^{+/+}::Lepr^{lox/lox}::Neo^{R}::Rosa26^{fs-tdTomato}$ mice, termed the control group (**Meng et al., 2016**). Immunohistochemistry and qPCR results showed the expression of tdTomato and deletion of *Lepr* in AgRP neurons 4 days after treatment of NB124, a synthetic nonsense-suppressor that can rapidly restore the function of Cre recombinase within the AgRP neurons (**Figure 1A–F** and **Figure 1—figure supplement 1**). Agrp-Lepr KO mice showed a significant increase in feeding and body weight one week after deletion of LepR signaling from AgRP neurons (**Figure 1G and H**). As a recapitulation of the severe hyperphagia caused by CRISPR-mediated LepR deletion (**Xu et al., 2018**), our Agrp-Lepr KO mice exhibited 44.5% increase in feeding as compared to the control mice. Furthermore, along with the increase of body weight and food intake, Agrp-Lepr KO mice exhibited impaired glucose tolerance (**Figure 1I**). Leptin induces phosphorylation of signal transducer and activator of transcription 3 (pSTAT3). pSTAT3 staining in the ARC showed enhanced extent in control mice but not in Agrp-Lepr KO mice 1 hr after leptin treatment in overnight fasted mice (**Figure 1—figure supplement 2**), indicating the role of leptin in the ARC was ablated. No significant changes of pSTAT3 level were observed in the DMH and VMH of control and Agrp-Lepr KO mice (**Figure 1—figure supplement 3**). Intriguingly, chronic infusion of bicuculline (Bic, 1 ng, a $GABA_A$ receptor antagonist) into the 3rd ventricle abolished the hyperphagia responses in Agrp-Lepr KO mice (**Figure 1J**), suggesting that facilitating the post-synaptic $GABA_A$ signaling to the AgRP neural circuit is important in regulating leptin-mediated feeding. The feeding efficiency (body weight gain [mg]/kilocalorie consumed) was significantly higher in Agrp-Lepr KO mice but was rescued by infusion of Bic (**Figure 1K**). The expansion of white adipose tissue (WAT) was observed in Agrp-Lepr KO mice (**Figure 1L**). We observed a significant decrease in respiratory quotient (RQ) from the Agrp-Lepr KO mice that can be further normalized by infusion of $GABA_A$ antagonist (**Figure 1M**). In line with other studies, these results suggest that GABA is a crucial signaling molecule by which AgRP neurons control adiposity and nutrient utilization (**Wu et al., 2009**; **Wu et al., 2008a**; **Tong et al., 2008**). The control and Agrp-Lepr KO mice were also analyzed for their glucose tolerance. Ablation of LepR in the AgRP neurons impaired glucose tolerance, which was fully restored by infusion of $GABA_A$ antagonist (**Figure 1N**). These results indicate that leptin regulates feeding and energy metabolism via post-synaptic $GABA_A$ signaling from AgRP neurons.

To identify the key downstream targets of AgRP neurons that contribute to leptin-mediated responses, we examined the *Npas4* expression in potential AgRP target neurons. Traditional immediate early genes are regulated by neuromodulators via cAMP, neurotrophins, and other paracrine factors, and their kinetics are relatively slow. In contrast, *Npas4* has a more dynamic response to activity-dependent signaling via $Ca^{2+}$, and it can be bidirectionally regulated by either excitatory or inhibitory input (**Shepard et al., 2019**). The qPCR results showed that rapid deletion of LepR signaling from AgRP neurons resulted in a significant decrease of *Npas4* abundance within the DMH (**Figure 2A**). To identify the functional relevance of each downstream target to leptin signaling, we administered DT into neonatal $Agrp^{DTR}$ mice to ablate all AgRP neurons and examined the neural activity of their downstream targets in response to leptin (**Figure 2B**; **Luquet et al., 2007**; **Zimmer et al., 2019**). Without ablation of AgRP neurons, leptin induced robust Fos expression in almost all major targets of AgRP neurons (**Wu et al., 2009**; **Wu et al., 2012**; **Wu et al., 2008b**). However, neonatal ablation of AgRP neurons significantly diminished the leptin-mediated Fos induction within post-synaptic neurons residing in the DMH (**Figure 2C–F** and **Figure 2—figure supplement 1**). These results suggest that the DMH could be the key downstream target that critically contributes to the regulation of leptin-mediated appetitive and metabolic responses. To further visualize the neural circuit from AgRP neurons to DMH, we injected a retrograde herpes simplex virus (HSV) encoding a red fluorescent protein (*HSV-hEf1α-mCherry*) into the DMH of $Npy^{GFP}$ mice (**Figure 2G**). Both HSV-based retrograde tracing and cholera toxin subunit B (CTB)-based retrograde neuroanatomical tracing from the DMH showed that ~17% (CTB tracing) or 39% (HSV tracing) of AgRP neurons project to the DMH (**Figure 2H–L** and **Figure 2—figure supplement 2**).

To establish a role of the DMH within the leptin-responsive AgRP neural circuit, we combined the transsynaptic tracer (WGA-ZsGreen) and whole-cell, patch-clamp recordings to facilitate electrophysiological analysis of neurons in the DMH that are innervated by AgRP neurons (**Figure 3A and**

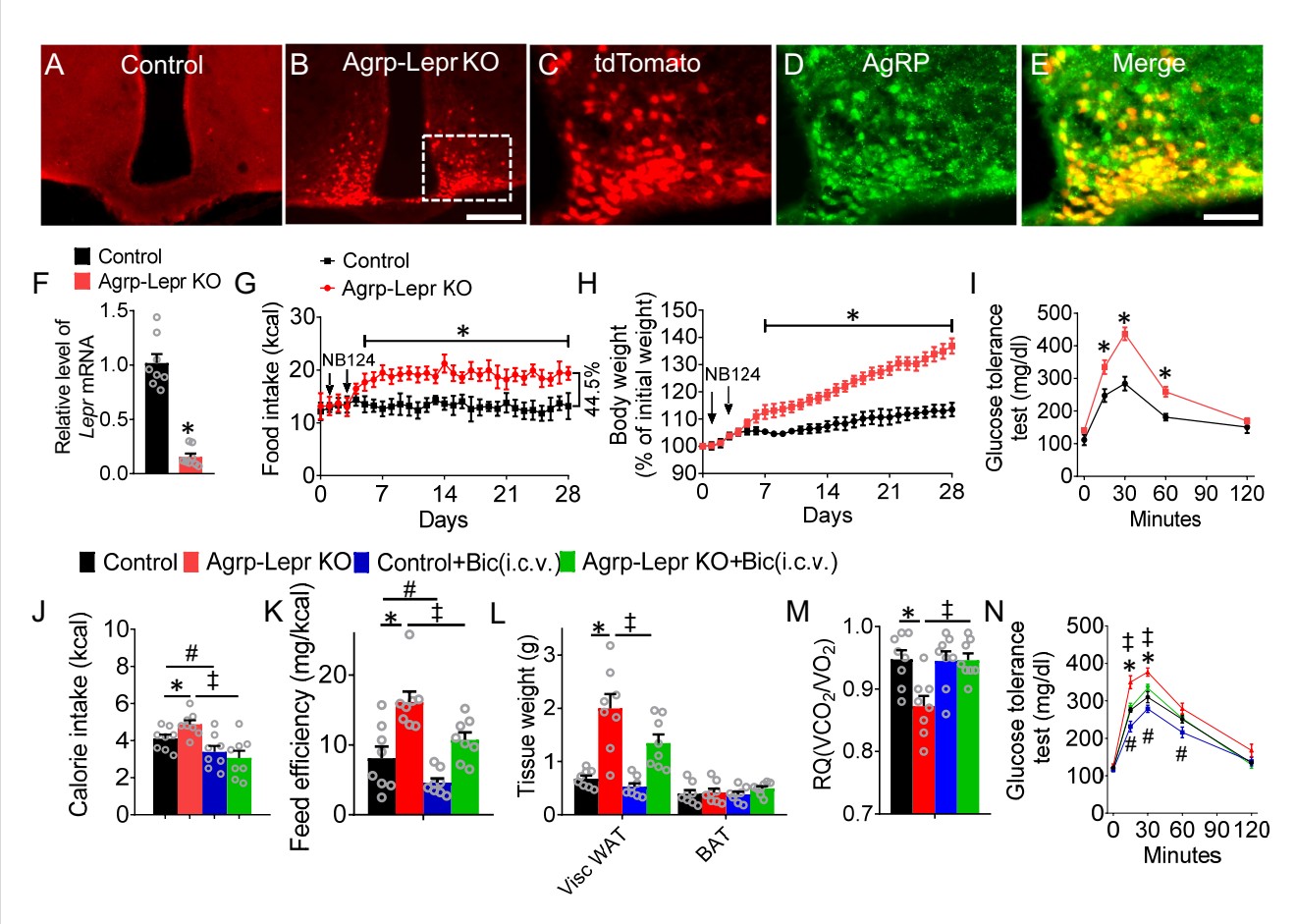

**Figure 1.** Suppressing GABA$_A$ signaling reverses feeding and metabolic dysfunction induced by leptin deficiency in AgRP neurons. (**A,B**) Expression of tdTomato in the arcuate nucleus with injection of NB124 into the 3$^{rd}$ ventricle for both *Agrp*$^{+/+}$*::Lepr*$^{lox/lox}$*::Neo*$^R$*::Rosa26*$^{fs-tdTomato}$ mice (control, (**A**)) and *Agrp*$^{nsCre/+}$*::Lepr*$^{lox/lox}$*::Neo*$^R$*::Rosa26*$^{fs-tdTomato}$ mice (Agrp-Lepr KO, (**B**)). Scale bar in *B* for *A* and *B*, 200 µm. (**C–E**) Colocalization of tdTomato (**C**) and anti-AgRP (**D**) in ARC. Scale bar in **E** for **C–E**, 100 µm. (**F**) Real-time qPCR analysis of *Lepr* transcript levels within the AgRP neurons of the control and Agrp-Lepr KO mice. (n=8 per group; *p<0.05). (**G–I**) Daily calorie intake (**G**) and body weight (**H**) of the control and *Agrp::Lepr*$^{KO}$ mice. The GTT (**I**) was tested on the day 21 after injection of NB124. (n=8 per group; *p<0.05). (**J**) The 4-hr food intake by the control and Agrp-Lepr KO mice after chronic infusion of Bic (1 ng) into the 3rd ventricle for 4 weeks. (**K**) Feeding efficiency is presented as mg of body weight gain/kcal consumed in the mice as described in **J**. (**L**) Weight of adipose tissues in the mice as described in **J** (Visc WAT: visceral white adipose tissue; BAT: brown adipose tissue). (**M**) Average value for RQ (defined as ratio of V$_{CO2}$/V$_{O2}$) tested on Day 28 in the mice as described in **J**. (**N**) The GTT was performed in the mice as described in **J**. (n=8 per group in **J–N**; *p<0.05 between Control and Agrp-Lepr KO, #p<0.05 between Control and Control +Bic(i.c.v.), ‡p<0.05 between Agrp-Lepr KO and Agrp-Lepr KO +Bic(i.c.v.)). Error bars represent mean ± SEM. unpaired two-tailed t test in **F**; one-way ANOVA and followed by Tukey comparisons test in **J–M**; two-way ANOVA and followed by Bonferroni comparisons test in **G–I**, and **N**.

The online version of this article includes the following source data and figure supplement(s) for figure 1:

**Source data 1.** The original data of body weight, food intake, and GTT for Figure 1.

**Figure supplement 1.** Representative images of the arcuate nucleus with injection of vehicle into the 3$^{rd}$ ventricle for both *Agrp*$^{+/+}$*::Lepr*$^{lox/lox}$*::Neo*$^R$*::Rosa26*$^{fs-tdTomato}$ mouse (**A**) and *Agrp*$^{nsCre/+}$*::Lepr*$^{lox/lox}$*::Neo*$^R$*::Rosa26*$^{fs-tdTomato}$ mouse (**B**).

**Figure supplement 2.** Representative images of pSTAT3 immunofluorescent staining in the ARC of fasted control and Agrp-Lepr KO mice with vehicle (**A–F**) or leptin i.c.v injections (**G–L**).

**Figure supplement 3.** Representative images of pSTAT3 immunofluorescent staining in the DMH and VMH of fasted control and Agrp-Lepr KO mice with vehicle (**A–B, E–F**) or leptin i.c.v injections (**C–D, G–H**).

*B*; *Han et al., 2021b*; *Han et al., 2021a*; *Xu et al., 2017*; *Xia et al., 2021*). RNA in situ hybridization confirmed that ~78% of AgRP-targeted neurons in the DMH express melanocortin receptor 4 (MC4R) (*Figure 3C* and *Figure 3—figure supplement 1*). To test whether this connectivity was monosynaptic, we perfused tetrodotoxin (TTX) and 4-aminopyridine (4-AP) into the bath to remove any network

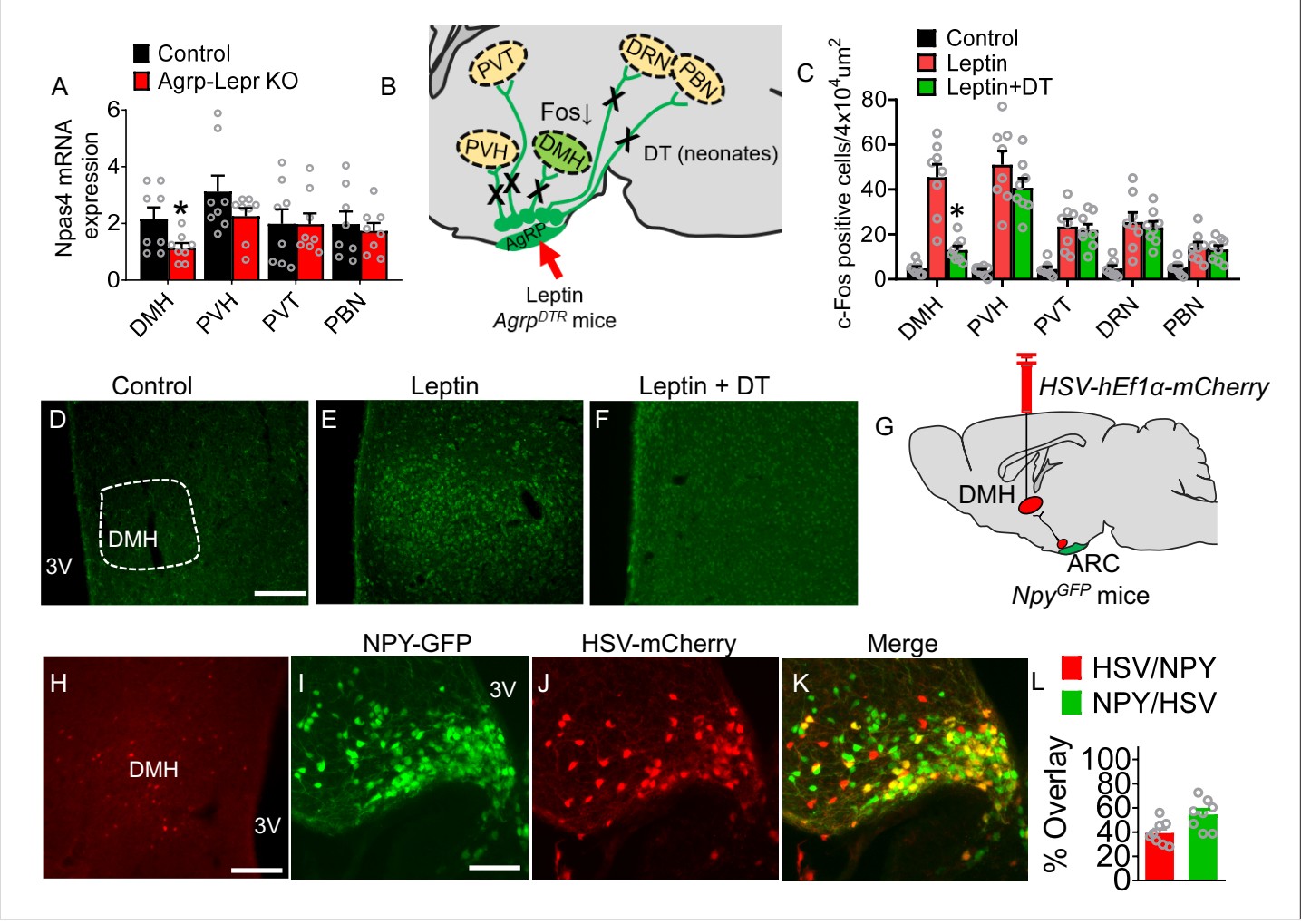

**Figure 2.** Identification of downstream targets of the leptin-responsive AgRP neurons. (**A**) Real-time qPCR analysis of transcript levels of *Npas4* expressed in the DMH, PVH, PVT, and PBN in the control and Agrp-Lepr KO mice. (n=8 per group; *p<0.05). (**B**) The schematic diagram showing Fos activities in downstream targets of AgRP neurons after treatment of leptin combined with or without neonatal ablation of AgRP neurons. (**C–F**) Expression profiling of Fos (green) in the DMH under vehicle (**D**), leptin (i.p.) (**E**), leptin (i.p.) and DT (s.c. neonates) (**F**). Scale bar in **D** for **D–F**, 200 μm. (n=8 per group; *p<0.05). (**G**) Diagram shows retrograde tracing of AgRP[ARC→DMH] neurons by injection of *HSV-hEf1α-mCherry* into the DMH of *Npy[GFP]* mice. (**H–K**) The fluorescence in the DMH (**H**) and ARC (I–K, yellow, DMH-projecting AgRP neurons). Scale bar in **H**, 200 μm; scale bar in **I** for (**I–K**), 100 μm. (**L**) Quantification of overlap of *HSV-hEf1α-mCherry* and NPY-GFP. (n=8 per group). Error bars represent mean ± SEM; unpaired two-tailed t test in **A**; one-way ANOVA and followed by Tukey comparisons test in **C**.

The online version of this article includes the following source data and figure supplement(s) for figure 2:

**Source data 1.** The histological data for Figure 2.

**Figure supplement 1.** Expression profiling of Fos (green) in the PVH, PVT, DRN, and PBN under vehicle (**A**), leptin (i.p.) (**B**), leptin (i.p.) and DT (s.c. neonates) (**C**).

**Figure supplement 2.** Retrograde tracing of AgRP[ARC→DMH] neurons by injection of CTB into the DMH of *Npy[GFP]* mice.

activity. We observed that inhibitory postsynaptic currents (IPSCs) in the DMH neurons triggered by photostimulation of ChR2-expressing axonal terminals of AgRP neurons were fully blocked by Bic (*Figure 3D and E*), confirming that these terminals were releasing GABA. In the presence of DNQX (a competitive AMPA/kainate receptor antagonist), AP5 (a selective NMDA receptor antagonist) and Bic, photostimulation of the ChR2-expressing AgRP terminals resulted in inhibition of action potentials in postsynaptic ZsGreen-positive neurons of the DMH in a reversible manner with significant reduction of firing rate and resting membrane potential (*Figure 3F–I*). We further examined the effect of leptin on the firing of postsynaptic ZsGreen-positive neurons in the DMH. We found that systemic treatment of leptin significantly enhanced neural activities of DMH neurons (*Figure 3J and K*).

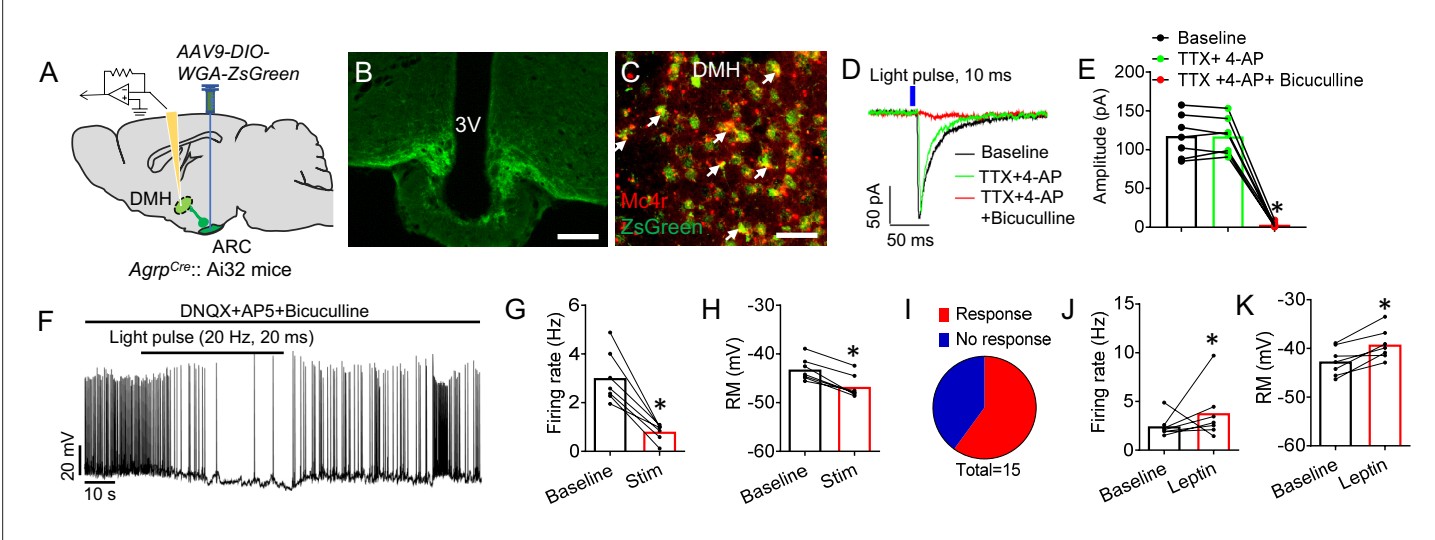

**Figure 3.** Leptin-responsive AgRP neurons directly innervate DMH neurons. (**A**) Diagram of transsynaptic tracing downstream targets from AgRP neurons. *AAV9-DIO-WGA-ZsGreen*, a Cre-dependent tracer that labels neurons with synaptic connections, was injected into the ARC of *Agrp^Cre*::Ai32 mice. Ai32 mice express ChR2/EYFP fusion protein conditioned Cre-mediated recombination. (**B**) ChR2-EYFP-labeled AgRP neurons in the ARC. Scale bar, 200 μm. (**C**) Double-labeling of MC4R neurons with transsynaptic WGA-ZsGreen in the DMH (**C**), green, *ZsGreen* RNA in situ; red, *Mc4r* RNA in situ. Scale bar, 100 μm. (**D**) The response of IPSCs recorded from an ZsGreen+ neuron upon photostimulation of AgRP terminals within the DMH (10ms pulse) with a pretreatment of vehicle or TTX +4 AP, and in slices pretreated with either TTX +4 AP or TTX +4-AP+Bic. (**E**) Statistical amplitude analysis of IPSCs from neurons recorded in **D**. (n=7 neurons; *p<0.05 between TTX +4 AP and TTX +4-AP+Bic). (**F**) Representative spikes of ZsGreen-labeled DMH neurons before and after blue light pulses (20ms/pulse, 20 Hz) shined onto AgRP axonal fibers. (**G–I**) Firing frequency (**G**), resting membrane (**H**), and statistical analysis (**I**) of neurons recorded in **F** (n=7 neurons; *p<0.05). (**J,K**) Firing frequency (**J**), and resting membrane (**K**) of ZsGreen-labeled DMH neurons with or without leptin i.p. treatment. (n=7 neurons; *p<0.05). Error bars represent mean ± SEM. paired two-tailed t test in **G**,**H**, **J**, and **K**; one-way ANOVA and followed by Tukey comparisons test in **E**.

The online version of this article includes the following source data and figure supplement(s) for figure 3:

**Source data 1.** The original data of firing rate and rest membrane potential from in vitro electrophysiological recording in Figure 3.

**Figure supplement 1.** Colocolization of ZsGreen and Mc4r in the DMH.

To examine the role of the AgRP→DMH circuit in the regulation of feeding and glucose, we performed optogenetic manipulation in *Agrp^Cre::Roas26^fs-ChR2* (Ai32) mice where ChR2-EYFP was selectively expressed in AgRP neurons and axonal terminals (*Han et al., 2021b*; *Xia et al., 2021*). Photostimulation of AgRP fibers in the DMH promoted feeding and impaired glucose tolerance (*Figure 4A and B*). We found that about 13% of DMH neurons could be sensitive to the activation of AgRP neurons (*Figure 4—figure supplement 1*). We did not observe significant changes of Fos expression in the ARC after activation of AgRP fiber in the DMH, which excludes the potential effects of activating AgRP neurons (*Figure 4—figure supplement 1D, H and J*). In contrast, photostimulation of postsynaptic MC4R^DMH neurons decreased feeding and improved glucose tolerance (*Figure 4C and D*). Bilateral infusion of Bic (4 ng) into the DMH of Agrp-Lepr KO mice with micro-osmotic pumps resulted in reduced feeding and a significant reduction of body weight, coupled with improved glucose tolerance (*Figure 4E–G*). Blockage of GABA_A receptor in the DMH significantly rescued the feeding efficiency, expansion of WAT, and the RQ profiles in Agrp-Lepr KO mice (*Figure 4H–K*). The effect of drug diffusion was examined by Fos expression in the DMH and the neighboring brain regions, showing no significant difference of Fos expression in these regions. (*Figure 4—figure supplement 2*). These results suggest that GABA_A receptor signaling in the DMH is critical for the feeding, body weight, and glucose tolerance regulated by LepR in AgRP neurons.

To better understand the physiological responses of these GABAergic neurons in the context of feeding and glucose regulation, we employed an in vivo opto-tetrode system to reveal the dynamic activities of MC4R^DMH neurons (*Han et al., 2021a*). With the injection of *AAV2-DIO-ChR2-GFP* into the DMH of *Mc4r^Cre* mice, we could record the activity of DMH neurons and identify those that express MC4R by their short-latency response to photostimulation as well as by the identical waveforms of evoked and spontaneous spikes (*Han et al., 2021a*). We investigated the activities of MC4R^DMH

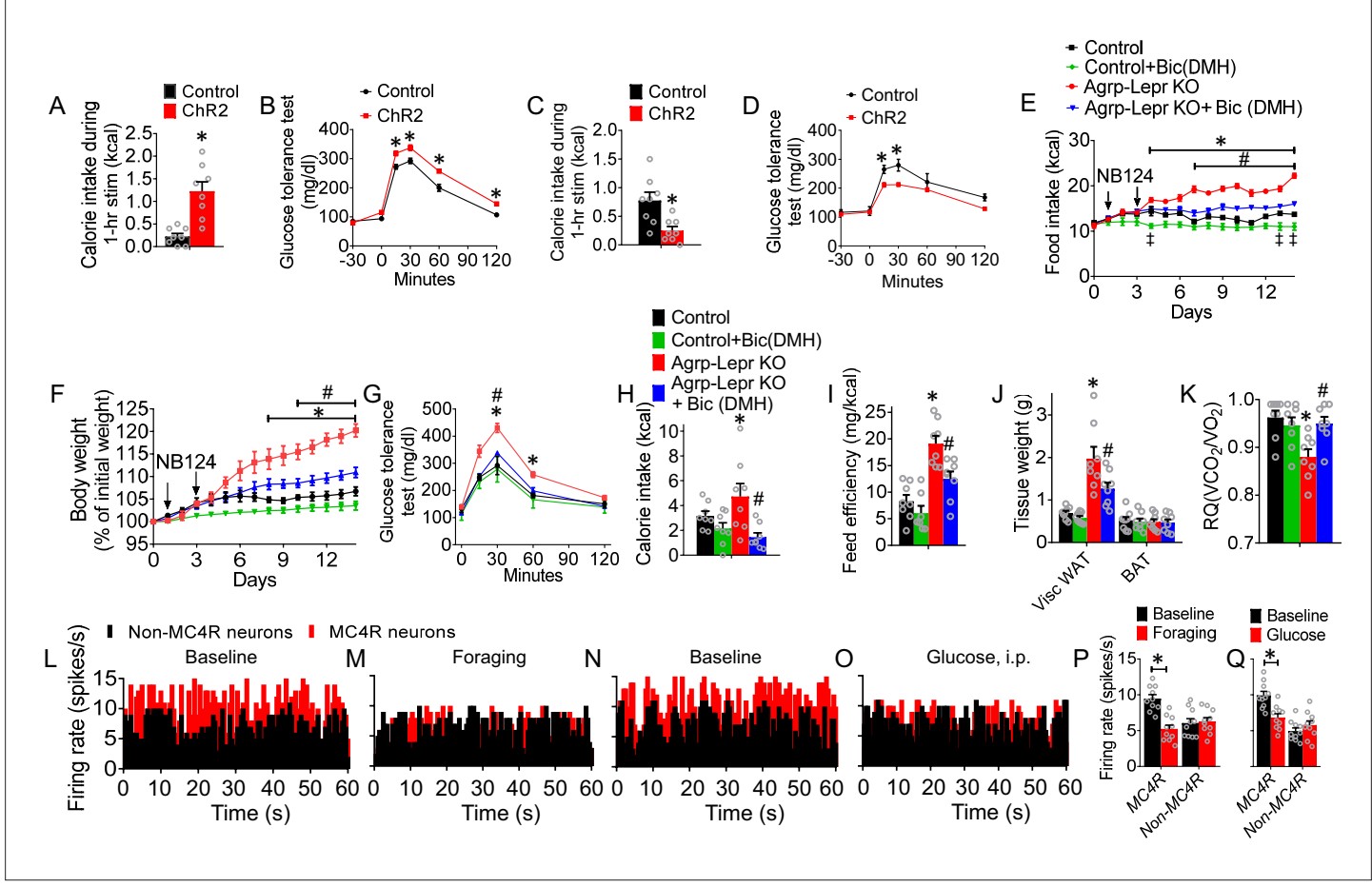

**Figure 4.** The leptin-responsive AgRP-DMH circuit regulates feeding, body weight, and glucose tolerance. (**A,B**) 1-hr food intake (**A**) and GTT (**B**) after photostimulation of the AgRP→DMH circuit. (n=8 per group; *p<0.05). (**C,D**) 1-hr refeeding test (**C**) and GTT (**D**) after photostimulation of the MC4R[DMH] neurons. (n=8 per group; *p<0.05). (**E–G**) Daily food intake (**E**), body weight (**F**) and GTT (**G**) in control mice, Agrp-Lepr KO mice with Bic or vehicle injected into the DMH. The saline or Bic (4 ng; 0.25 µl/hr) was chronically infused into the DMH via the osmotic minipump for 14 days. (n=8 per group; *p<0.05 between Control and Agrp-Lepr KO, #p<0.05 between Agrp-Lepr KO and Agrp-Lepr KO +Bic(DMH), ‡p<0.05 between Control and Control +Bic(DMH)). (**H**) 4-hr food intake in the mice described in E-G on day 14. (**I**) Feed efficiency in the mice described in **E–G**. (**J**) Weight of adipose tissues in the mice described in **E–G** on day 14. (**K**) Average value for RQ ($V_{CO_2}/V_{O_2}$) in the mice described in **E–G** on day 14. (n=8 per group in **H–K**; *p<0.05 between Control and Agrp-Lepr KO, #p<0.05 between Agrp-Lepr KO and Agrp-Lepr KO +Bic(DMH)). (**L–O**) Firing rate of representative MC4R[DMH] neurons and non-MC4R[DMH] neurons when mice under baseline (satiation) (**L, N**), foraging (prior to meal initiation) (**M**), and glucose i.p. injection (well-fed state) (**O**). The statistical analysis of the firing rate under baseline, foraging (**P**), and glucose treatment (**Q**) were calculated. (n=10 neurons from 3 mice in each group; *p<0.05). Error bars represent mean ± SEM. unpaired two-tailed t test in **A**, **C**, **P** and **Q**; one-way ANOVA and followed by Tukey comparisons test in **H–K**; two-way ANOVA and followed by Bonferroni comparisons test in **B** and **D–G**.

The online version of this article includes the following source data and figure supplement(s) for figure 4:

**Source data 1.** The original data of body weight, food intake, GTT, and firing rate of MC4R neurons in the DMH for Figure 4.

**Figure supplement 1.** Immunostaining of Fos and WGA-ZsGreen in the DMH of *Agrp[Cre]::Roas26[fs-ChR2]* mice with (**E–G**) or without (**A–C**) optogenetic activation of AgRP fiber in the DMH.

**Figure supplement 2.** The Fos expression in the DMH and other brain regions after infusing Bic into DMH.

neurons and non-MC4R[DMH] neurons under chow diet and hyperglycemia conditions. A total of 16 MC4R[DMH] neurons and 12 non-MC4R[DMH] neurons were identified through optogenetic-invoked spikes. The results showed that 10 out of 16 identified MC4R neurons showed decreased firing rate during foraging (prior to meal initiation), with an average reduction of firing rate from 9.8 Hz to 5.1 Hz (***Figure 4L, M and P***). We also observed 10 out of 16 MC4R neurons that responded to the enhancement of blood glucose with a reduction of firing rate from 10.0 Hz to 7.1 Hz (***Figure 4N, O and Q***). These results are consistent with MC4R[DMH] neurons' mediation of foraging and glucose tolerance.

Our results showed that GABA$_A$ receptors in the DMH are involved in the regulation of feeding and glucose tolerance; thus, we attempted to identify the key GABA$_A$ receptor subunits that are functionally relevant to glucose and feeding regulation. Results of qPCR analysis showed that, among all major regulatory α subunits, the transcript level of *Gabra3* (encoding GABA$_A$ receptor α3 subunits) in the DMH neurons of Agrp-Lepr KO mice was the highest (**Figure 5A**). Immunostaining confirmed that α3-GABA$_A$ was abundantly expressed within the DMH (**Figure 5B**). Our transsynaptic tracing study further confirmed that the α3-GABA$_A$ signaling was highly co-localized within the post-synaptic targets of the AgRP→DMH neural circuit (**Figure 5C–E**). To understand the functional roles of α3-GABA$_A$ signaling within DMH neurons for leptin-mediated feeding and glucose, *Mc4r^{Cre}::Rosa26^{fs-Cas9}* mice were injected with *AAV9-Gabra3^{sgRNA}-tdTomato* into the DMH. We found that knockout of *Gabra3* signaling in the MC4R$^{DMH}$ neurons reduced feeding and body weight while glucose tolerance was significantly improved (**Figure 5F–H** and **Figure 5—figure supplement 1**). Deficiency in α3-GABA$_A$ signaling also significantly decreased feeding efficiency and WAT (**Figure 5I and J**). A gain-of-function study showed that overexpression of α3-GABA$_A$ within the same MC4R$^{DMH}$ neurons manifested opposite phenotypes, including moderately increased feeding and body weight and significantly increased feeding efficiency and WAT adiposity (**Figure 5F–J**). Importantly, overexpression of α3-GABA$_A$ in the DMH neurons blunted the actions of leptin on feeding, body weight, glucose tolerance, feeding efficiency, and WAT adiposity (**Figure 5K–O**). To determine the role of α3-GABA$_A$ receptor signaling from the MC4R$^{DMH}$ neurons in the regulation of feeding and glucose tolerance by activation of AgRP-DMH circuit, *Npy^{Flp}::Mc4^{Cre}* mice with were injected with *AAV9-fDIO-ChR2-EYFP* into the ARC, *AAV9-DIO-Cas9-mCitrine* and *AAV9-Gabra3^{sgRNA}-tdTomato* into the DMH of followed with implantation of optic fiber in the DMH. We found that knockout of *Gabra3* signaling in the MC4R$^{DMH}$ neurons abolished the enhancement of feeding and impairment of glucose tolerance during activation of AgRP-DMH circuit (**Figure 5—figure supplement 2**). These results suggest the AgRP-DMH circuit regulates feeding and glucose tolerance depending on the α3-GABA$_A$ receptor signaling in the MC4R$^{DMH}$ neurons. We further examined the effects of ablating α3-GABA$_A$ in the DMH on feeding and body weight in diet-induced obese mice. We found that genetic deletion of *Gabra3* signaling in the MC4R$^{DMH}$ neurons led to a decrease in daily food consumption associated with a moderate weight loss and that these knockout mice showed reduced refeeding responses (**Figure 5P–R**). These results suggest that the MC4R$^{DMH}$ neurons play a significant role in control of glucose tolerance and feeding through the α3-GABA$_A$ signaling.

## Discussion

Leptin exerts its behavioral and metabolic effects by modulating signaling of AgRP neurons that are susceptible to obesity-induced leptin resistance. In this report, we explored the neural circuit and transmitter-signaling mechanisms underlying leptin-mediated feeding and energy metabolism (**Figure 5S**). We identified and characterized a unique GABAergic neural circuit with functional significance: the neural circuit from LepR-expressing AgRP neurons to α3-GABA$_A$ receptor expressing DMH neurons plays a critical role in the control of feeding, body weight and glucose tolerance through α3-GABA$_A$ receptor signaling. This leptin-responsive neural circuit plays a fundamental role in the regulation of hyperphagia and metabolism in obesity, which suggests manipulation of the neural circuit pharmacologically could lead to novel obesity therapeutics.

This study utilized a newly established inducible-knockout strategy to achieve rapid, post-developmental, targeted inactivation of LepR signaling, which precisely illuminates the pathophysiological roles of central leptin signaling on the control of nutrient partitioning and feeding efficiency. Compared with the mild effects of the non-inducible manipulation of brain LepR signaling, our Agrp-Lepr KO model showed a robust disruption in various feeding and metabolism parameters, culminating in severe obesity, which was previously achieved by global genetic deletion or the CRISPR–Cas9 technique (**Xu et al., 2018**; **van de Wall et al., 2008**; **Gonçalves et al., 2014**; **Egan et al., 2017**). Our technique has many attractive features, such as the large repertoire of conditional mouse lines, transient and non-BBB-crossing inducer, and easy combination with numerous viral tools, that can be applied to many neurological and endocrine questions.

We applied HSV to map the AgRP-DMH circuit. HSV-hEF1α has been extensively used to retrogradely transport from the peripheral nerve terminals to the central nervous system through axonal transport (**Fang et al., 2018**; **Tan et al., 2016**; **Eagle et al., 2020**; **Yamaguchi et al., 2020**; **Garcia**

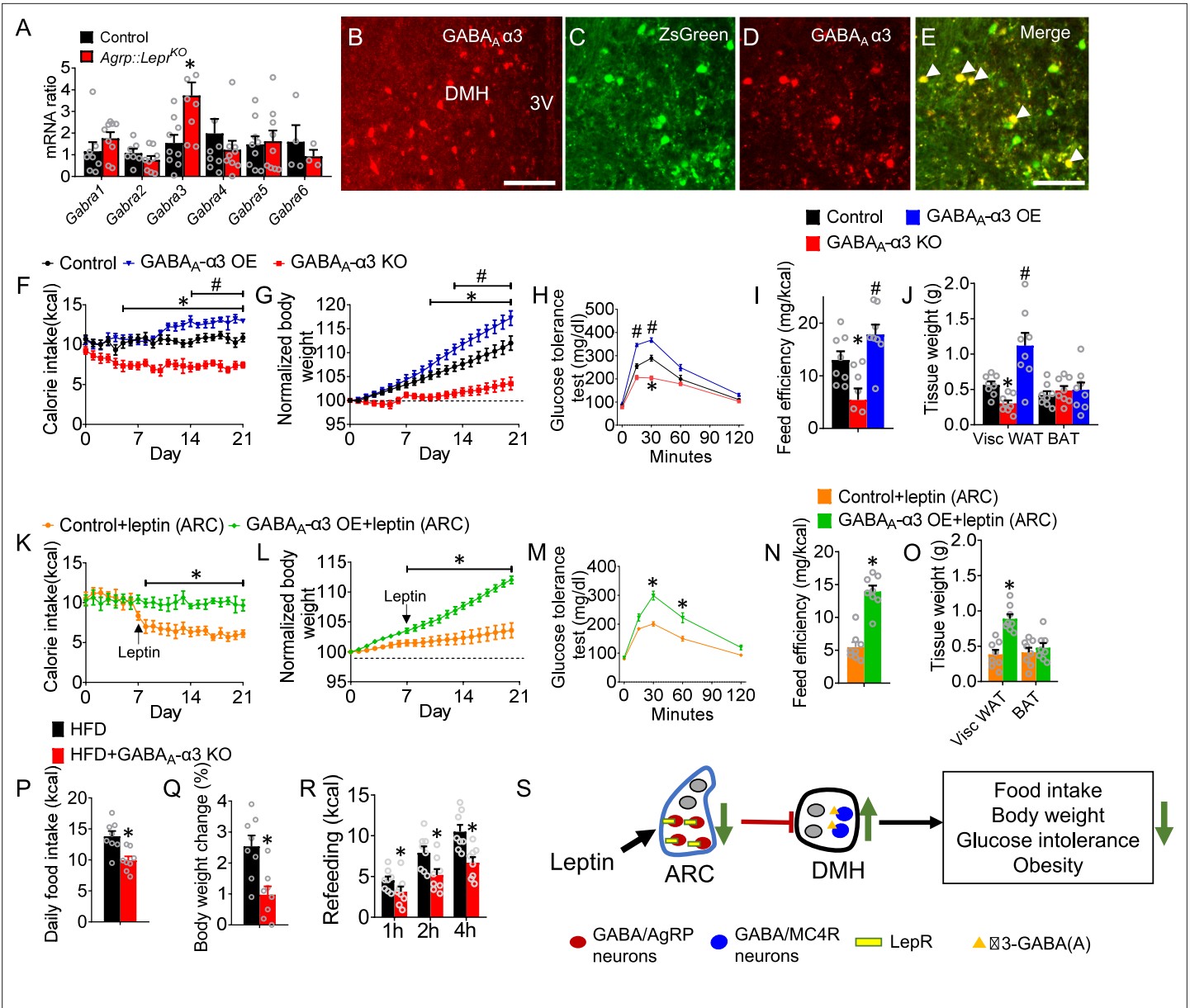

**Figure 5.** The α3-GABA$_A$ receptor signaling within the MC4R$^{DMH}$ mediates feeding and glucose tolerance. (**A**) Expression level of *Gabra1- Gabra6* in the DMH neurons of control and Agrp-Lepr KO mice. (**B**) Immunostaining showing the expression of GABA$_A$ α3 in the DMH. Scale bar, 100 μm. (**C–E**) Colocalization of GABA$_A$ α3 and transsynaptically labeled neurons by WGA-ZsGreen in the DMH after injection of *AAV9-DIO-WGA-ZsGreen* into the ARC of *Agrp$^{Cre}$* mice. Scale bar in **E** for **C–E**, 50 μm. (**F–H**) The food intake (**F**), body weight (**G**), and GTT (**H**) were performed in *Mc4r$^{Cre}$::Rosa26$^{fs-Cas9}$* mice with a bilateral injection of vehicle, *AAV9-Gabra3$^{sgRNA}$-tdTomato* (GABA$_A$-α3 KO), or *AAV9-DIO-Gabra3$^{cDNA}$-EYFP* (GABA$_A$-α3 OE) into the DMH. (**I**) Feeding efficiency in the mice described in **F–H**. (**J**) Weight of adipose tissues in the mice described in **F–H**. n=8 per group in **F–J**; *p<0.05 between Control and GABA$_A$-α3 KO, #p<0.05 between Control and GABA$_A$-α3 OE. (**K–O**) The food intake (**K**), body weight (**L**), GTT (**M**), feeding efficiency (**N**), and weight of adipose tissues (**O**) were performed in control mice and GABA$_A$-α3 OE mice followed by the microinjection of leptin into ARC. (n=8 per group in K-O; *p<0.05 between Control +leptin and GABA$_A$-α3 OE +leptin). (**P–R**) The 24-hr food intake (**P**), body weight change (**Q**), and refeeding test (**R**) were performed in HFD-treated with or without knockout of GABA$_A$-α3. (n=8 per group; *p<0.05). Error bars represent mean ± SEM. unpaired two-tailed t test in **A** and **N–R**; one-way ANOVA and followed by Tukey comparisons test in **I** and **J**; two-way ANOVA and followed by Bonferroni comparisons test in **F–H** and **K–M**. (**S**) Diagram showing a leptin regulated GABAergic neural circuit reverses obesity. The GABAergic AgRP$^{LepR}$→DMH circuit plays a critical role in control of leptin-mediated food intake, body weight, and glucose tolerance through the α3-GABA$_A$ signaling within the MC4R$^{DMH}$ neurons.

The online version of this article includes the following source data and figure supplement(s) for figure 5:

**Source data 1.** The original data of body weight, food intake, and GTT for Figure 5.

*Figure 5 continued on next page*

*Figure 5 continued*

**Figure supplement 1.** Quantification of *gabra3* transcription levels in the DMH by qPCR 21 days after injection of *AAV9-Gabra3$^{sgRNA}$-tdTomato* into DMH of *Mc4r$^{Cre}$::Rosa26$^{fs-Cas9}$* mice (**A**) or injection of *AAV9-DIO-Gabra3$^{cDNA}$-EYFP* into DMH of *Mc4r$^{Cre}$* mice (**B**).

**Figure supplement 2.** The AgRP-DMH circuit regulates feeding and glucose tolerance depending on the α3-GABA$_A$ receptor signaling in the MC4R$^{DMH}$ neurons.

*et al., 2021*; *Gremel et al., 2016*; *Valentinova et al., 2019*; *Marcinkiewcz et al., 2016*), while some applications take advantage of the anterograde capacity of other different strains for example HSV-H129 (*Azevedo et al., 2019*; *Lo and Anderson, 2011*). To further verify the AgRP-DMH circuit, we applied CTB which is capable of being taken up by neurons and transported in a retrograde direction towards the cell body. Consistent with HSV tracing results, CTB tracing strategy also showed the connection between AgRP neurons and DMH.

We applied pharmacological approaches to block GABA$_A$ signaling within the AgRP-DMH circuit, which can blunt the hyperphagia responses and improve glucose intolerance in Agrp-Lepr KO mice. These results suggest that the GABA$_A$ signaling system within the AgRP-DMH circuit exerts rapid actions on feeding behaviors and glucose metabolism. This shows the consistency that GABA in the AgRP neurons is required for the stimulation of feeding (*Krashes et al., 2013*). Furthermore, we observed decreased food intake and weight loss as a result of chronic blockage of GABA$_A$ signaling in the DMH, indicating that GABA$_A$ signaling within the AgRP-DMH circuit may also participate in the chronic regulation of feeding and body weight. On the other hand, some studies showed that AgRP neurons can chronically regulate feeding through the mechanism of long-term suppression over the downstream brain targets independent of GABA$_A$ signaling (*Krashes et al., 2013*; *Wang et al., 2021*). Together, we suggest that the GABA signaling in the AgRP neurons may exert chronic effects on appetite and weight control via a distinct neural pathway from the co-released AgRP and NPY.

The profiling assay using neonatal ablation of AgRP neurons revealed key neural populations responding to leptin. Among various downstream targets of the AgRP circuit, the DMH neurons are the most sensitive to AgRP-dependent leptin signaling. Acute stimulation of post-synaptic MC4R neurons within the DMH blunted glucose tolerance and feeding.

The GABA$_A$ subunits in the hypothalamus are involved in the regulation of feeding. We employed the CRISPR-Cas9, gene-editing method to comprehensively evaluate the function of GABA$_A$ subunits α3 within a leptin-responsive neural circuit. Genetic deletion of α3 in the MC4R neurons in the DMH suppressed feeding and glucose tolerance. These studies demonstrate that the GABA$_A$ subunits α3 within the secondary structure of the GABAergic neural circuit play an important role in the control of leptin-mediated hypometabolism and obesity.

Utilizing the in vivo optrode recording system, we performed stable intracellular recordings from MC4R neurons in the DMH in intact and awake mice, allowing us to study complex behaviors and physiological responses. Here, we focused on the effects of MC4R neurons in the DMH on sensing glucose levels. There are several brain regions such as ARC, LH, PVH, VMH, and DMH that could sense peripheral glucose levels including glucose-excited and glucose-inhibited neurons (*Routh et al., 2014*; *Claret et al., 2007*; *He et al., 2020*; *Huang et al., 2022*; *Burdakov et al., 2005*). Our results showed that 62.5% of MC4R neurons in the DMH neurons are relevant to hyperglycemia due to their suppressed neural activity associated with high glucose levels. Others showed either unresponsive or opposite phenotypes. Overall, the effects of glucose on MC4R neurons in the DMH suggest that glucose is a complex and dynamic metabolic signal that interacts with multiple neural circuits to regulate glucose homeostasis.

In conclusion, LepR in the AgRP neurons and the associated neural circuit and receptor are the primary mediators of leptin action in feeding and metabolic homeostasis. We suggest that new therapeutics selectively targeting α3-GABA$_A$ associated signaling components within this GABAergic neural circuit would prevent obesity.

## Materials and methods

### Animals

All animal care and experimental procedures were approved by the Institutional Animal Care and Use Committees at Baylor College of Medicine. Mice used for data collection were at least eight weeks

old males; and kept in temperature- and humidity-controlled rooms, in a 12/12 hr light/dark cycle, with lights on from 6:00 AM-6:00 PM. Health status was normal for all animals. All experiments are independently performed.

Genetic mouse models: $Agrp^{DTR/+}$ mice (**Luquet et al., 2005**), $Agrp^{Cre}$ mice (**Tong et al., 2008**), $Agrp^{nsCre}$ mice (**Meng et al., 2016**), Lepr$^{lox/lox}$ mice (**Cohen et al., 2001**), Neo$^R$ mice (**Aubrecht et al., 2011**), Ai32 mice (**Madisen et al., 2012**), $Npy^{GFP}$ mice (**van den Pol et al., 2009**), $Rosa26^{fs-tdTomato}$ mice (**Madisen et al., 2012**), $Mc4r^{Cre}$ mice (**Krashes et al., 2014**), $Npy^{Flp}$ mice (**Daigle et al., 2018**), and $Rosa26^{fs-Cas9}$ mice (**Platt et al., 2014**) were produced as described previously. All mice are on a C57Bl/6 background with at least 8 generations backcrossed.

## Stereotaxic surgery

All the mice with brain surgery were performed with the same pre-operative and post-operative care as described previously (**Han et al., 2021a**). Briefly, animals were received analgesia buprenorphine (s.c.) 1 hr prior to the start of anesthesia then anesthetized with isoflurane and placed on a stereotaxic frame (David Kopf, Tujunga, CA). The local anesthetics were applied before making an incision. All surgeries were performed on a heating pad and allowed to recover in a heating cage.

To perform the viral injection, virus was loaded into a needle (Hamilton Small Hub RN 33 G, Reno, NV) connected with a 10 µL syringe (Hamilton 700 Microliter, Reno, NV). Injections were performed with an Ultra MicroPump (World Precision Instruments, Sarasota, FL) and Micro4 Controller (Heidenhain Corporation, Schaumburg, IL), at a rate of 0.1 µL/min. For AAV a total 0.3 µl volume was delivered into brain regions. For HSV a total 0.2 µl was delivered into the DMH The relevant stereotaxic coordinates for the injections are described in the following according to a standardized atlas of the mouse brain (Franklin and Paxinos, third edition, 2007).

For viral injection, the coordinate of ARC is AP = –2.06 mm, ML = ±0.25 mm, DV = –5.9 mm; the coordinate of DMH is AP = –1.94 mm, ML = ±0.3 mm, DV = –5.3 mm.

For cannulation of optical fiber, the coordinate of DMH is AP –1.94 mm, ML +0.3 mm, DV –4.9 mm.

The virus $AAV2$-$EF1a$-$DIO$-$hChR2(E123T/T159C)$-$GFP$ was from UNC; the $HSV$-$hEf1\alpha$-$mCherry$ were from Harvard; the plasmids p$AAV$-$Gabra3^{sgRNA}$-$tdTomato$, p$AAV$-$DIO$-$Gabra3^{cDNA}$-$EYFP$, p$AAV$-$DIO$-$Cas9$-$mCitrine$ were synthesized by our lab. All noncommercial viruses including $AAV9$-$DIO$-$WGA$-$zsGreen$ (**Xia et al., 2021**), $AAV9$-$fDIO$-$ChR2$-$EYFP$ were packaged by Optogenetics and Viral Design/Expression Core at Baylor College of Medicine. All of viruses were diluted to a final working titer with no less than $2\times10^{12}$ viral genomes per ml.

## Ablation of AgRP neurons in neonates

To ablate AgRP neurons, the newborn $Agrp^{DTR/+}$ pups were either injected with diphtheria toxin (75 ng in 20 µl saline, s.c.) or saline. After weaning the genotype was determined by PCR. The 28 days after DT injection the mice were treated with leptin (4.0 mg/kg, i.p. twice per day) for 3 days. The mice were euthanized, and the brain was harvested. The whole brain was sectioned to 20 µm thickness by a microtome (ThermoFisher Scientific, Waltham, MA). Immunostaining for Fos (1:1500 dilution; EMD Millipore, Burlington, MA) was performed. Fluorescent images of Cy2-labeled Fos +neurons in brain regions were obtained by an Axio Observer microscope (Zeiss, Thornwood, NY) and further analyzed using ImageJ software (NIH).

## Optogenetics

For in vivo optogenetic stimulation, the optic fiber was assembled as described following the protocol (**Zhang et al., 2007**). To perform the photostimulation the optical fiber was connected to spectralynx (Neuralynx, Inc, USA) through a patch cable. For the food intake measurement, the blue light was shed into the DMH at 20 Hz with 10ms pulse for 1 hr. The power of laser (0.5 mW) was calculated by optical power meter (PM100D, Thorlabs, Newton, NJ) before each experiment.

For in vitro optogenetics, the optic fiber was assembled as described following the protocol (**Zhang et al., 2007**). The blue light was controlled by a pulse stimulator. The blue light pulses (20 Hz, 10ms/pulse) were shed onto the ChR2-expressing AgRP axonal fibers within the DMH. The power of the laser (0.5 mW) was measured by a power meter (PM100D, Thorlabs, Newton, NJ) before experiments.

## Drug administration

To general administration of leptin, the mice received i.p. injection of leptin at a dose of 4.0 mg/kg. The mice were sacrificed for Fos examination 2 hr later or for in vitro electrophysiology 30 min later.

For infusing bicuculline (Bic) into the 3rd ventricle, a circular craniotomy (diameter 0.5 mm) was drilled at the locations of DMH. The guide cannula (26 gauge, Plastics One, Roanoke, VA) was installed on the holder and guided into the target brain region. To deliver drug the internal cannula was inserted onto the top of the guide cannula, extended below the guide cannula 0.5 mm, and Bic (1 ng; Sigma-Aldrich, St Louis, MO) was applied daily for 4 weeks.

To chronically infuse Bic to the DMH Alzet micro-osmotic pumps (model 1002, Durect, Cupertino, CA, USA) loaded with 100 µl of Bic (4 ng; 0.25 µl/hr) were implanted subcutaneously on the back of mice for 14 days. The cannula was implanted into DMH with the coordinate (AP –1.94 mm, ML +0.3 mm, DV –4.8 mm), and the Alzet minipumps were connected to the cannulas by tubing (PE60; Stoelting, Wood Dale, IL).

For infusing leptin into the ARC, the guide cannula (26 gauge, Plastics One, Roanoke, VA) was implanted in the locations of ARC (AP –20.6 mm, ML +0.25 mm, DV –5.4 mm). Leptin (2.5 µg/side) was infused.

To infuse NB124 to into the 3rd ventricle, the mice were anesthetized and NB124 (*Clemmensen et al., 2020*) (two injections of 0.4 mg/side, 2 days apart; Calbiochem, San Diego, CA) was administrated.

## Food intake

The feeding test was performed 3 weeks after virus injection. For the acute feeding studies in optogenetic experiments, regular food intake was measured (from the start of the 'lights off' cycle, 6 pm - 7 pm) one hour during and after 1 hr photostimulation with chow diet (5V5R, LabDiet, St. Louis, MO). For the refeeding test in optogenetic experiments, food intake (1 hr) was measured from 12 pm to 1 pm with chow diet after 18 hr fasting from 6 pm to 12 pm. The regular food intake (4 hr) of chow diet in well-fed mice was monitored from 6 pm to 10 pm. For the chronic feeding studies, food intake as well as body weight was measured daily between 9 am and 10 am up to 4 weeks. For the refeeding test with high-fat diet (HFD, 60% kcals from fat, Bio-Serv), food intake (1 hr, 2 hr, and 4 hr) was monitored from 12 pm to 4 pm after 18 hr fasting from 6 pm to 12 pm.

## Glucose tolerance test (GTT)

The GTT was performed as described (*Clemmensen et al., 2020*). Briefly, mice were fasted overnight for 16 hr, a blood sample was taken, and then the mice were injected with D-glucose (1 g/kg, i.p.), and blood was drawn from the tail vein at 0, 15, 30, 60, and 120 min later. Blood glucose levels were determined with a FreeStyle Lite glucometer (Abbott Laboratories).

## Energy expenditure

The $O_2$ consumption and $CO_2$ production were monitored by Comprehensive Lab Animal Monitoring System (CLAMS; Columbus Instruments, Columbus, OH) (*Clemmensen et al., 2020*). The respiratory quotient (RQ, defined as ratio of $V_{CO2}/V_{O2}$) was calculated that provides an indication of the nature of the substrate being used by an organism (i.e. RQ = 1 for glucose utilization; RQ = 0.7 for lipid utilization). Mice were acclimatized in the chambers for 48 hr prior to data collection.

## In vitro electrophysiology

Mice were deeply anesthetized with isoflurane and transcardially perfused with a modified ice-cold sucrose-based cutting solution (pH 7.3) containing 10 mM NaCl, 25 mM $NaHCO_3$, 195 mM Sucrose, 5 mM Glucose, 2.5 mM KCl, 1.25 mM $NaH_2PO_4$, 2 mM Na-Pyruvate, 0.5 mM $CaCl_2$, and 7 mM $MgCl_2$, bubbled continuously with 95% $O_2$ and 5% $CO_2$. The mice were then decapitated, and the entire brain was removed and immediately submerged in the cutting solution. The brains of adult mice were sectioned in coronal plane with thickness of 250–300 µm by a Microm HM 650 V vibratome (ThermoFisher Scientific, Waltham, MA). The brain slices were kept in artificial cerebrospinal fluid (aCSF) as described recently (*He et al., 2016*). Slices containing the DMH were recovered for 1 hr at 34 °C and then maintained at room temperature in artificial cerebrospinal fluid (aCSF, pH 7.3) containing 126 mM NaCl, 2.5 mM KCl, 2.4 mM $CaCl_2$, 1.2 mM $NaH_2PO_4$, 1.2 mM $MgCl_2$, 11.1 mM glucose, and

21.4 mM NaHCO$_3$ saturated with 95% O$_2$ and 5% CO$_2$ before recording. Slices were transferred to a recording chamber and allowed to equilibrate for at least 10 min before recording. The slices were superfused at 34 °C in oxygenated aCSF at a flow rate of 1.8–2 ml/min. ZsGreen-labeled neurons in the DMH were visualized using epifluorescence and IR-DIC imaging on an upright microscope (Eclipse FN-1, Nikon) equipped with a moveable stage (MP-285, Sutter Instrument). Patch pipettes with resistances of 3–5 MΩ were filled with intracellular solution (pH 7.3) containing 128 mM K-Gluconate, 10 mM KCl, 10 mM HEPES, 0.1 mM EGTA, 2 mM MgCl2, 0.05 mM Na-GTP and 0.05 mM Mg-ATP. Recordings were made using a MultiClamp 700B amplifier (Axon Instrument), sampled using Digidata 1440 A and analyzed offline with pClamp 10.3 software (Axon Instruments). Series resistance was monitored during the recording, and the values were generally <10 MΩ and were not compensated. The liquid junction potential was +12.5 mV and was corrected after the experiment. Data were excluded if the series resistance increased dramatically during the experiment or without overshoot for action potential. Currents were amplified, filtered at 1 kHz, and digitized at 20 kHz. Current clamp was engaged to test neural firing frequency and resting membrane potential (Vm) at the baseline. The aCSF solution contained 1 µM tetrodotoxin (TTX) and a cocktail of fast synaptic inhibitors, AP-5 (30 µM; an NMDA receptor antagonist) and DNQX (30 µM; an AMPA receptor antagonist) to block the majority of presynaptic inputs. For the light evoked inhibitory postsynaptic current (IPSC) recordings, the internal recording solution contained: 125 mM CsCH$_3$SO$_3$; 10 mM CsCl; 5 mM NaCl; 2 mM MgCl$_2$; 1 mM EGTA; 10 mM HEPES; 5 mM (Mg)ATP; 0.3 mM (Na)GTP (pH 7.3 with NaOH). IPSC within the DMH neurons was measured in the current clamp mode in the presence of 1 µM TTX and 4-AP with or without 50 µM bicuculline.

## In vivo tetrode recording

We used the microdrives (Neuralynx) that enabled photostimulation and recording neural activities simultaneously (*Anikeeva et al., 2011*; *Liu et al., 2010*). The microdrives were loaded with one optic fiber in the center and 7 nichrome tetrodes consisting of 4 thin wires twined together (STABLOHM 675, California Fine Wire Co., Grover Beach, CA). The optic fiber was surrounded by a bundle of tetrodes that are positioning 0.1 mm below optic fiber. Tetrode were plated with gold to reduce impedance to 0.3–0.4 M (tested at 1 kHz). The microdrive was implanted into the DMH in the *Mc4r$^{Cre}$* mice with injection of *AAV2-EF1a-DIO-hChR2(E123T/T159C)-GFP* within the DMH. Then the mouse was connected to a 32-channel preamplifier headstage (Neuralynx). All signals recorded from each tetrode were amplified, filtered between 0.3 kHz and 6 kHz, and digitized at 32 kHz by Neuralynx. The local field potentials were amplified and filtered between 0.1 Hz and 1 kHz. The tetrodes were slowly lowered in quarter-turns of a screw on the microdrive with about 60 µm per step. Spikes were sorted by Offline Sorter software (Plexon). Units were separated by the T-Distribution E-M method, and cross-correlation and autocorrelation analyses were used to confirm unit separation. Clustered waveforms were subsequently analyzed by using NeuroExplorer (Nex Technologies, Colorado Springs, CO) and MATLAB (MathWorks, Natick, MA). The firing rates were presented with spikes per bin with 1 s interval or spikes per second. The ChR2 +neurons were identified by the short latencies of evoked spikes accurately following high-frequency photostimulation, as well as the identical waveforms of evoked and spontaneous spikes (*Kvitsiani et al., 2013*).

## Real-time qPCR

Sorted cells were immediately lysed by RLT buffer from the RNeasy Plus Micro kit (Qiagen). Total mRNA was subsequently extracted and purified based on the manufacturer's suggested protocol (Qiagen). For some experiments, total RNA was extracted from fresh arcuate nucleus by TRIzol (Invitrogen) per the manufacturer's instructions. The purified RNA was quantified by One-drop spectrophotometer (ThermoFisher Scientific, Waltham, MA), and mRNA was reverse-transcribed by using the SuperScript II kit (Invitrogen) per the manufacturer's suggested protocol. qPCR was performed in the Bio-Rad CFX96 Real-Time PCR system. Relative abundance of *Lepr*, $\alpha$ subunits of GABA$_A$ receptor transcripts were determined by using the 2$-\Delta\Delta$Ct method and normalized to Gapdh, a housekeeping gene. The Real-time PCR was performed using TaqMan gene expression assay (IDT, Coralville, IA) for *Lepr* (Mm.PT.58.33275723), *Gabra1* (Mm.PT.51.8672637), *Gabra2* (Mm.PT.51.12495699), *Gabra3* (Mm.PT.51.12076884), *Gabra4* (Mm.PT.51.5838999), *Gabra5* (Mm.PT.51.13771433), *Gabra6* (Mm.PT.51.10447632), *Gapdh* (Mm.PT.39a.1).

## Histology

Immunostaining was performed as described with modification (*Clemmensen et al., 2020*). Mice were killed and perfused transcardially with ice-cold PBS buffer (pH 7.4) containing 3% (wt/vol) para-formaldehyde (Alfa Aesar, Ward Hill, MA) and 1% glutaraldehyde (Sigma, St. Louis, MO). Brains were collected and postfixed overnight under 4 °C in a fixation buffer containing 3% paraformaldehyde. Free-floating sections (25 µm) were cut by a microtome (ThermoFisher Scientific, Waltham, MA) and then blocked with 5% (wt/vol) normal donkey serum in 0.1% Triton X-100 (TBST buffer, pH 7.2) for overnight. Goat anti-AgRP (1:500 dilution; Santa Cruz Biotech, Dallas, TX), rabbit anti-Fos (1:1500 dilution; EMD Millipore, Burlington, MA), pSTAT3 (1:400 dilution; Cell signaling, Danvers, MA), GABRA3 (1:500 dilution; Abcam, Waltham, MA) were applied to the sections per different experiments for overnight incubation under 4 °C, followed by 4×15 min rinses in the TBST buffer. Finally, sections were incubated with Alex Fluor Cy2-conjugated secondary antibody (1:1000 dilution; Jackson Immunolab, West Grove, PA) or Alex Fluor Cy3-conjugated secondary antibody (1:1000 dilution; Jackson Immunolab, West Grove, PA) for 2 hr at room temperature, followed by 4×15 min rinses in TBST buffer. For mounted sections, fluorescent images were captured by a digital camera mounted on an Axio Observer microscope (Zeiss, Thornwood, NY).

Dual mRNA in situ hybridization (ISH) was performed on 25-µm-thick DMH brain sections cut from fresh-frozen brain from age-matched controls. We generated a digoxigenin (DIG)-labeled mRNA antisense probes against Mc4r and fluorescein (FITC)-labeled mRNA against Zsgreen using reverse-transcribed mouse cDNA as a template and an RNA DIG or FITC-labeling kits from Roche (Sigma). Primer and probe sequences for the Mc4r and Zsgreen probe are available in Allen Brain Atlas (http://www.brain-map.org) and https://www.genepaint.org. ISH was performed by the RNA In Situ Hybridization Core at Baylor College of Medicine using an automated robotic platform as previously described (*Yaylaoglu et al., 2005*) with modifications of the protocol for double ISH. After the described washes and blocking steps the DIG-labeled probes were visualized using tyramide-Cy3 Plus (1/50 dilution, 15-min incubation, Perkin Elmer). After washing in TNT the remaining HRP-activity was quenched by a 10-min incubation in 0.2 M HCl. The sections were then washed in TNT, blocked in TNB for 15 min before a 30-min room temperature incubation with HRP-labled sheep anti-FITC antibody (1/500 in TNB, Roche). After washing in TNT the FITC-labeled probe was visualized using tyramide-FITC Plus (1/50 dilution, 15-min incubation, Perkin Elmer). Following washing in TNT the slides were stained with DAPI (invitrogen), washed again, removed from the machine and mounted in ProLong Diamond (Invitrogen).

## Statistical analyses

Data were analyzed by unpaired t test, paired t test, one-way or two-way ANOVA with the post hoc as appropriate. Statistical analyses were performed using Prism software (GraphPad Software, San Diego, CA). Results were considered significantly different at $p < 0.05$. All data are presented as mean ± S.E.M.

## Acknowledgements

We acknowledge support from the Metabolomics Core and the Mouse Metabolism Core at Baylor College of Medicine, the Optogenetics and Viral Design/Expression Core at Baylor College of Medicine. We thank Cecilia Ljungberg with the RNA In Situ Hybridization Core facility at Baylor College of Medicine for the technical support on the in situ hybridization. This work was supported by NIH Shared Instrumentation grant S10 OD016167, P30 DK056338 (to Cecilia Ljungberg); R01DK109194, R01DK131596 (to Q Wu), the Pew Charitable Trust awards 0026188 (to Q Wu), USDA/CRIS grants 3092-5-001-059 (to Q Wu), the USDA/ARS Faculty Start-up grants (to Q Wu), USDA/CRIS grants 3092-5-001-059 (to Y Xu); Q Wu is the Pew Scholar of Biomedical Sciences and the Kavli Scholar.

## Additional information

### Funding

| Funder | Grant reference number | Author |
|---|---|---|
| NIH Office of the Director | R01DK109194 | Qi Wu |
| NIH Office of the Director | R01DK131596 | Qi Wu |
| USDA | 3092-5-001-059 | Qi Wu<br>Yong Xu |

The funders had no role in study design, data collection and interpretation, or the decision to submit the work for publication.

### Author contributions

Yong Han, Data curation, Formal analysis, Investigation, Methodology, Writing - original draft, Writing - review and editing; Yang He, Data curation, Formal analysis, Investigation, Methodology; Lauren Harris, Methodology; Yong Xu, Resources, Funding acquisition; Qi Wu, Conceptualization, Resources, Data curation, Supervision, Funding acquisition, Validation, Investigation, Visualization, Methodology, Writing - original draft, Project administration, Writing - review and editing

### Author ORCIDs

Yong Han (ID) http://orcid.org/0000-0003-0768-046X
Qi Wu (ID) http://orcid.org/0000-0003-0303-4065

### Ethics

All animal care and experimental procedures were approved by the Institutional Animal Care and Use Committees (Protocol # 6598) at Baylor College of Medicine. All the mice with brain surgery were performed with pre-operative and post-operative care to minimize suffering.

### Decision letter and Author response

Decision letter https://doi.org/10.7554/eLife.82649.sa1
Author response https://doi.org/10.7554/eLife.82649.sa2

## Additional files

### Supplementary files

• Transparent reporting form

### Data availability

All data generated or analyzed during this study are included in the manuscript and supporting file.

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
