## [Editor Report]

Leptin is a fat-derived hormone that curbs appetite, and mutation of leptin causes obesity and diabetes. This manuscript investigates leptin-responsive neural circuits, revealing a key inhibitory connection from leptin-sensitive neurons in the arcuate nucleus of the hypothalamus (AGRP neurons) to neurons in the dorsomedial hypothalamus. Toggling this inhibitory connection impacted leptin effects on feeding and metabolism.

---

## [Decision Letter]

**Decision letter after peer review:**

[Editors’ note: the authors submitted for reconsideration following the decision after peer review. What follows is the decision letter after the first round of review.]

Thank you for submitting the paper "Identification of a GABAergic neural circuit governing leptin-mediated obesity" for consideration by *eLife*. Your article has been reviewed by 3 peer reviewers, and the evaluation has been overseen by a Reviewing Editor and a Senior Editor. The reviewers have opted to remain anonymous.

Comments to the Authors:

We are sorry to say that, after consultation with the reviewers, we have decided that this work will not be considered further for publication by *eLife* at this time.

The reviewers thought the work was of potential interest, but that substantial additional experiments were required to validate claims. It was discussed that if comments can be addressed in full, a re-submission would be encouraged, but that the revisions were substantial enough that they would not fit in the timeline for an invited *eLife* revision.

*Reviewer #1 (Recommendations for the authors):*

1. It is confusing why DMH neuron activity is suppressed in response to food, and the authors should explain this counterintuitive observation in the results or discussion. Food intake should decrease AGRP neuron activity, which should decrease inhibitory inputs to DMH neurons and thus increase neuronal firing unless I am missing something.

2. The bicuculline experiments are not so clean; in Figure 1 panels, % changes generally seem similar between the control and test groups. It seems possible/likely that effects are not epistatic, but rather involve two separate pathways whose effects are additive. This is a major challenge to the conclusions of the paper and should be discussed. Also, bicuculline applications in control mice are missing in all relevant panels of Figure 5.

3. In figure 3C, what is the percentage of WGA+ neurons that express MC4R, and what is the percentage of MC4R neurons that express WGA? Mc4r mice are used later in the text, but it is not clear that they provide a suitable tool. On a related note, for Figure 5, what percentage of Mc4R neurons are sensitive to AGRP neuron activation?

4. An inhibitory connection between AGRP neurons and Mc4R neurons is proposed, but there is no discussion of AGRP itself, which should provide a complementary pathway for inhibition over a longer time scale.

5. In several cases, the language is confusing as is and additional information should be provided.

5a. The title should be edited, as leptin does not mediate obesity; do the authors mean '…governing leptin mutation-induced obesity' or '…governing obesity induced by leptin mutation'?

5b. 'GABA signaling' in lines 181-193 should be replaced with 'GABA synthesis' or 'GABA production'. GABA signaling confusingly implies that receptor manipulations were made- it took me a few reads before I realized what the authors' manipulations actually were and that there was a 3rd negative feedback loop at play.

5c. On line 150, the transsynaptic tracer is called 'ZsGreen' but should be called 'WGA-ZsGreen' for precision; WGA is the key component that enables tracing.

5d. It should be clarified whether the control mice in Figure 1 also received NB124.

5e. Text in the discussion about GABAA signaling and feeding regulation is redundant with the introduction. It only needs to be discussed once.

5f. The abstract and text should be edited for grammar. For example, there are several 'the's missing in some places and extra 'the's in others.

5g. In reference to Figure 1L, the expansion of WAT is claimed to account for the 'entire' body weight change, but 1L shows a 1g increase, which is much less than the overall body weight increase in 1H.

5h. The authors should clarify why the baselines are different between 5B and 5D in the legend. I assume animals were in different fed/fasted states but this should be explicitly stated.

5i. The positions of Figure 5 panels should be adjusted so that it is clear that the legends above I apply to K and L as well.

*Reviewer #2 (Recommendations for the authors):*

The new genetic system for ablation of Lepr from Agrp neurons is poorly explained, and no data are shown to validate its effects.

Figure 2 – HSV has been shown to trace anterogradely, yet it is used here as a retrograde tracer without validation.

Figure 3 – This new tracing system seems interesting, but is not explained or validated. The amount of overlap between zsGreen and MC4R is not quantified – how equivalent are these cell types? The sample size for the ephys experiments (especially in panels G-K) is woefully small.

Are MC4R and AgRP target neurons in the DMH identical?

Figure 4 – This system also seems like it should ablate GABA signaling in Agrp neurons. There is no validation of the system.

Figure 5 – There is no validation of the KO or OE. The manipulation of the GABAergic system here is very general, and there is no analysis of the effects of these manipulations on the action of AgRP neurons.

*Reviewer #3 (Recommendations for the authors):*

1) The framework of the paper is misleading – they argue that (1) the ARC is the primary site of action for leptin on food intake and (2) that a lack of knowledge of the circuits mediating leptin's effect undermines its utility for treatment. However, this framework does not make sense with the current manuscript. First, many papers have highlighted roles for LepR signaling outside of the ARC and even outside of the hypothalamus for critical roles in feeding. Second, this work does not investigate leptin resistance which is at least in part a major contributor to the failure of leptin for treating obesity. While the manuscript provides insight into leptin receptor circuitry that mediates feeding, the current framework does not fit with the experiments and results.

2) The studies that utilize GABA receptor pharmacology are inconclusive given the doses and major effects that the antagonist has on its own. For example, in figure 1, bicuculline on its own has huge effects on food intake feed efficiency and glucose tolerance. Therefore, the data are hard to interpret. A dose of the antagonist that is ineffective on its own but blocks the effects of the KO is necessary to claim an interaction between the systems, especially since GABA is such an omnipresent neurotransmitter throughout the brain.

3) The experiments manipulating AgRP-DMH neurons lack proper controls.

– The DMH is not very far from the ARC. How did the authors verify that they did not stimulate ARC cell bodies with the laser light?

– Similarly, how did the authors verify that the infusion of bicuculline did not spread beyond DMH?

– In the experiment where GABA signaling in AgRP-DMH neurons was knocked out, how did the authors verify that these effects were specific to only the AgRP neurons that project to the DMH?

It is very important to verify selective manipulation of this pathway in order to make the claims that are made in the manuscript.

4) A few questions about the opto-tetrode recordings:

– 13 MC4R neurons and 9 non-MC4R neurons were recorded – this is a low number. How many mice were used for this study? Is this enough for a representative sample?

– I am confused why glucose seemingly inhibits MC4R neurons. If AgRP neurons inhibit MC4R, and glucose inhibits AgRP neurons, shouldn't they be activated?

5) Some of the histological data lack quantification.

6) Overall, the language throughout the manuscript is very operational, especially when describing the experiments. I would recommend conceptualizing the experiments, in addition to describing the tools (both within the text and in the figure legends) to improve flow and readability.

[Editors’ note: further revisions were suggested prior to acceptance, as described below.]

Thank you for resubmitting your work entitled "Identification of a GABAergic neural circuit governing leptin-mediated obesity" for further consideration by *eLife*. Your revised article has been evaluated by Ma-Li Wong (Senior Editor) and a Reviewing Editor.

The manuscript has been improved but there are some remaining issues that need to be addressed, as outlined below.

Both reviewers felt strongly that there are several claims that are not substantiated in the manuscript and that all remaining comments need to be addressed in full prior to re-evaluation. The full comments are detailed below.

*Reviewer #2 (Recommendations for the authors):*

The authors have added new data to the manuscript, including some important controls, and the manuscript is much improved.

Some issues remain, however:

Figure 1 (or appropriate supplement)- should show nsCre reporter activity in nsCre mice that have not received NB124.

Supplemental Figure 1- is pSTAT3 normal in other areas (e.g., the DMH, VMH)?

Figure 3- the biccuculine experiment still suffers from the unknown extent of diffusion. Perhaps staining for FOS might show the extent of diffusion (based on the absence of FOS).

Figure 4M-O- as noted previously, expressing Flp-dependent WGA-Cre in Npy neurons may also delete Gad1/2 in these neurons. It should also cause deletion in other regions that are targeted by Agrp neurons. Thus, this is not DMH-specific.

While the finding that deleting Gabra3 in the DMH blocks leptin-induced anorexia is useful, it would be better to show that this maneuver blocks the effects of deleting Lepr from Agrp neurons or some other Agrp neuron-specific manipulation.

Overall, while the manuscript remains interesting, the lack of specificity in some of the experiments does not allow the authors to cleanly conclude the model that they propose.

*Reviewer #3 (Recommendations for the authors):*

The authors addressed some of my previous concerns and have also provided several new verifications for techniques which improves confidence in the manuscript. Some concerns remain which are described below. Another major issue is that, unless I missed it, there is no methods section included in the revised manuscript, making it difficult to evaluate many of the new experiments.

Specific remaining issues:

– While the authors revised the manuscript using a lower dose of bicuculline, significant effects remain compared to controls for food intake, feed efficiency, and glucose tolerance, and therefore the original concerns remain for Figure 1. In Figure 4, analyses from repeated measures ANOVA were not reported and so it is unclear if there are differences between control and biculculline groups, and given the lack of methods here the experimental paradigm is unclear (e.g., what is a chronic, 5-min injection via osmotic minipump? – Figure 4 legend)?

– The fact that the HSV results yielded 2x the overlap as the CTB experiment raises concern about using it as an anterograde tracer, as previously noted by another reviewer. This should be discussed at the very least, or perhaps the HSV results should be omitted completely.

– The authors' editorial attempts to address concerns about site specificity are not very compelling and this is important as ARC and DMH are relatively near each other (and DMH is close to the ventricle – where infused substances could diffuse). Data demonstrating that optogenetic stimulation only activates AgRP neurons that project to DMH, and that infused substances do not reach 4V or ARC, would be helpful, and if this is not possible a discussion of these limitations is warranted.

– I am still confused by the recordings of MC4R neurons – where both foraging and hyperglycemia inhibit MC4R neurons. The authors cite one study suggesting that glucose activates AgRP neurons but do not discuss the many other studies that show the opposite. At the very least this should be discussed rather than the conclusion being that "these findings are consistent with MC4R-DMH neurons' mediation of food intake and blood glucose"

– I am also still confused with how the DMH GABA KO studies fit into the narrative of the paper. The effect size here is quite large (and it is an interesting finding), but the leptin signaling is systemic and not specific to AgRP neurons.

---

## [Author Response]

[Editors’ note: The authors appealed the original decision. What follows is the authors’ response to the first round of review.]

Essential revisions:Reviewer #1 (Recommendations for the authors):1. It is confusing why DMH neuron activity is suppressed in response to food, and the authors should explain this counterintuitive observation in the results or discussion. Food intake should decrease AGRP neuron activity, which should decrease inhibitory inputs to DMH neurons and thus increase neuronal firing unless I am missing something.

Thanks for your comments. We agree to the common view that AgRP neurons generally decrease their activities when food ingestion has been initiated or animals are sated. In this manuscript, however, we focus on the neural activities of AgRP neurons during the foraging stage in which AgRP neurons are known to be activated by hunger. Therefore, the neural firing of DMH neurons is subsequently suppressed as shown here (Figure 5E). To rule out any confusion, we updated the main text (lines 191), and legend (line 568-569).

2. The bicuculline experiments are not so clean; in Figure 1 panels, % changes generally seem similar between the control and test groups. It seems possible/likely that effects are not epistatic, but rather involve two separate pathways whose effects are additive. This is a major challenge to the conclusions of the paper and should be discussed. Also, bicuculline applications in control mice are missing in all relevant panels of Figure 5.

We appreciate these valuable comments. We performed a new experiment with lower dose of bicuculline to show the difference between the control and experimental groups (Figure 1J-N). Under a lower dose, bicuculline treatment in the DMH showed little effects on food intake, glucose tolerance, and energy metabolism on control group. But this treatment can abolish the hyperphagia responses in Agrp-Lepr KO mice. These observations strongly argue that bicuculline exerts the rescuing effect via direct action upon this neural circuit.

We also updated new Figure 4 with additional bicuculline applications in control groups.

3. In figure 3C, what is the percentage of WGA+ neurons that express MC4R, and what is the percentage of MC4R neurons that express WGA? Mc4r mice are used later in the text, but it is not clear that they provide a suitable tool. On a related note, for Figure 5, what percentage of Mc4R neurons are sensitive to AGRP neuron activation?

Thanks for your comments. We performed further histological analysis and updated the manuscript. We found that 78% WGA+ neurons within the DMH co-express MC4R, and that 55% MC4R neurons within the DMH co-express WGA (Figure S4G and lines 139). We photostimulated the AgRP fibers in the DMH in *Agrp-Cre::Ai32* mice, and performed c-Fos staining to examine the percentage of DMH neurons could be inactivated by photostimulation of AgRP neurons. We found there are ~13% of DMH neurons responded to the activation of AgRP (Figure S5)

4. An inhibitory connection between AGRP neurons and Mc4R neurons is proposed, but there is no discussion of AGRP itself, which should provide a complementary pathway for inhibition over a longer time scale.

We added more discussions (lines 244-257) to the manuscript accordingly.

5. In several cases, the language is confusing as is and additional information should be provided.5a. The title should be edited, as leptin does not mediate obesity; do the authors mean '…governing leptin mutation-induced obesity' or '…governing obesity induced by leptin mutation'?

Thanks for your comments. We modified the title in our revised manuscript per your suggestion.

5b. 'GABA signaling' in lines 181-193 should be replaced with 'GABA synthesis' or 'GABA production'. GABA signaling confusingly implies that receptor manipulations were made- it took me a few reads before I realized what the authors' manipulations actually were and that there was a 3rd negative feedback loop at play.

Thanks for your comments. We have modified the description (lines 170-180) as reviewer suggested.

5c. On line 150, the transsynaptic tracer is called 'ZsGreen' but should be called 'WGA-ZsGreen' for precision; WGA is the key component that enables tracing.

We corrected the error in our revised manuscript (line 136).

5d. It should be clarified whether the control mice in Figure 1 also received NB124.

We updated our revised manuscript (lines 490-492).

5e. Text in the discussion about GABAA signaling and feeding regulation is redundant with the introduction. It only needs to be discussed once.

Thanks for your comments. We updated it in our revised manuscript.

5f. The abstract and text should be edited for grammar. For example, there are several 'the's missing in some places and extra 'the's in others.

We corrected the error in our revised manuscript.

5g. In reference to Figure 1L, the expansion of WAT is claimed to account for the 'entire' body weight change, but 1L shows a 1g increase, which is much less than the overall body weight increase in 1H.

We updated our revised manuscript (lines 105-106) to describe the results more accurately.

5h. The authors should clarify why the baselines are different between 5B and 5D in the legend. I assume animals were in different fed/fasted states but this should be explicitly stated.

Thanks for your comments. The neural firing of MC4R neurons and non-MC4R neurons in the DMH from Figure 5B was recorded under hunger condition, and the neural firing from Figure 5D was recorded under well-fed and glucose treatment state. We updated the legend accordingly (lines 568-569).

5i. The positions of Figure 5 panels should be adjusted so that it is clear that the legends above I apply to K and L as well.

The current arrangement of Figure 5 has a better flow for the ephys recording assay, and genetic studies regarding the GABAA receptor subunits.

Reviewer #2 (Recommendations for the authors):The new genetic system for ablation of Lepr from Agrp neurons is poorly explained, and no data are shown to validate its effects.

We appreciate these critiques. We examined the pSTAT3 in the arcuate nucleus of fasted mice with or without leptin treatment. We found that pSTAT3-positive neurons were much more abundant upon leptin treatment in the control group. In contrast, we observed a significant decrease of pSTAT3 signals in Agrp-Lepr KO mice as compared to the control (Figure S1 and lines 96-100). These results suggest that our genetic KO model can efficiently and acutely disrupt the leptin receptor signaling within AgRP neurons.

Figure 2 – HSV has been shown to trace anterogradely, yet it is used here as a retrograde tracer without validation.

Thanks for your comments. Several previous studies established that HSV showed retrograding capacity (1-3). To further validate the AgRP-DMH neural circuit, we performed another retrograde tracing assay using CTB as the retrograde tracer. Consistent with the HSV results, the CTB tracing results indicated that AgRP neurons project to the DMH (Figure S3, lines 131134).

Figure 3 – This new tracing system seems interesting, but is not explained or validated. The amount of overlap between zsGreen and MC4R is not quantified – how equivalent are these cell types? The sample size for the ephys experiments (especially in panels G-K) is woefully small.

Thanks for your comments. We updated the manuscript with statistically quantified data (Figure S4 and lines 131-134) and added new data in the ephys recording.

Are MC4R and AgRP target neurons in the DMH identical?

We performed further histological analysis and updated the manuscript. We found that 78% WGA+ neurons within the DMH co-express MC4R, and that 55% MC4R neurons within the DMH co-express WGA (Figure S4). These results suggest that a large cohort of AgRP postsynaptic neurons are MC4R neurons.

Figure 4 – This system also seems like it should ablate GABA signaling in Agrp neurons. There is no validation of the system.

We performed the ablation of GABA synthesis in DMH neurons (not in AgRP neurons) that receive projection from AgRP neurons in Figure 4. We added validation data showing the KO of *Gad1*, *Gad2* and the decrease of GABA neurons in DMH (Figure S6 and S7).

Figure 5 – There is no validation of the KO or OE. The manipulation of the GABAergic system here is very general, and there is no analysis of the effects of these manipulations on the action of AgRP neurons.

We performed the qPCR of GABAA-α3 in KO and OE mice (Figure S8).

Reviewer #3 (Recommendations for the authors):1) The framework of the paper is misleading – they argue that (1) the ARC is the primary site of action for leptin on food intake and (2) that a lack of knowledge of the circuits mediating leptin's effect undermines its utility for treatment. However, this framework does not make sense with the current manuscript. First, many papers have highlighted roles for LepR signaling outside of the ARC and even outside of the hypothalamus for critical roles in feeding. Second, this work does not investigate leptin resistance which is at least in part a major contributor to the failure of leptin for treating obesity. While the manuscript provides insight into leptin receptor circuitry that mediates feeding, the current framework does not fit with the experiments and results.

We appreciate these valuable comments and critiques. Although the LepR signaling outside of the ARC and those extra-hypothalamus have been recently documented involved in various aspects of feeding and metabolism regulation, a recent genetic study established a pivotal role of the LepR within the AgRP neurons in appetite and obesity control as the AgRP LepR-KO model can near perfectly phenocopy *db/db* mice by some key parameters, such as hyperphagia, massive weight gain, and severe metabolic deficiency (4). Our current study provides the first-hand evidence in understanding the complex secondary structure of the leptin responsive neural circuit and signaling mechanism that had never been explored. Further study in deciphering the post-synaptic targets of non-ARC LepR-expressing neurons will provide a more comprehensive knowledge in the neural circuit control of the leptin signaling.

We agree to the reviewer that it is of high significance of understanding the neural mechanisms underlying leptin resistance for the treatment of obesity. However, our current study focuses, for the first time, on a neural circuit that governs the leptin-responsive feeding, weight control, and glucose metabolism under regulation of a specific post-synaptic GABAA signaling pathways with the effect against obesity. The current manuscript has been thoroughly revamped to fully support the proposed neural circuit framework as our initial attempt to understand the complex secondary structure of the leptin-responsive neural circuit in control of appetite and body weight. This study will undoubtedly justify further study of the leptin-mediated brain mechanism.

2) The studies that utilize GABA receptor pharmacology are inconclusive given the doses and major effects that the antagonist has on its own. For example, in figure 1, bicuculline on its own has huge effects on food intake feed efficiency and glucose tolerance. Therefore, the data are hard to interpret. A dose of the antagonist that is ineffective on its own but blocks the effects of the KO is necessary to claim an interaction between the systems, especially since GABA is such an omnipresent neurotransmitter throughout the brain.

We appreciate these valuable comments. We performed a new experiment with lower dose of bicuculline to show little effect on food intake, glucose tolerance, and energy metabolism on the knockout and control groups (Figure 1J-N). Notably, bicuculline at this lower dose, while showing little effect on food intake in the control group, can effectively abolish the hyperphagia responses in Agrp-Lepr KO mice. These observations strongly argue that bicuculline exerts the rescuing effect via direct action upon this neural circuit.

3) The experiments manipulating AgRP-DMH neurons lack proper controls.– The DMH is not very far from the ARC. How did the authors verify that they did not stimulate ARC cell bodies with the laser light?

Thanks for your comments. To avoid non-specific deep penetration of laser in the brain tissue we controlled the laser intensity by measuring the laser power from the tip of fiber. The power of laser is limited to 0.5mW that will only stimulate neurons within the DMH (~0.3mm below the optic fiber (5), also refer to: https://nicneuro.net/optogenetics-depth-calculator/) so that ensuring no direct impact upon AgRP cell bodies in the ARC.

– Similarly, how did the authors verify that the infusion of bicuculline did not spread beyond DMH?

We verified the placement of cannula within the DMH region upon completion of all behavioral assays and excluded the data from individuals with off-targeted infusion.

– In the experiment where GABA signaling in AgRP-DMH neurons was knocked out, how did the authors verify that these effects were specific to only the AgRP neurons that project to the DMH?It is very important to verify selective manipulation of this pathway in order to make the claims that are made in the manuscript.

In the experiment of Figure 4M-O, we performed the genetic ablation of GABA synthesis in the DMH GABAergic neurons (not AgRP neurons) that receive projection from AgRP neurons. Our viral knockout strategy permits specific targeting the DMH neurons that are transsynaptically connected with AgRP neurons, and only GABA synthesis in these Cre+ DMH neurons was disrupted. We added validation data showing the KO of *Gad1*, *Gad2* and the decrease of GABA neurons in DMH (Figure S6 and S7).

4) A few questions about the opto-tetrode recordings:– 13 MC4R neurons and 9 non-MC4R neurons were recorded – this is a low number. How many mice were used for this study? Is this enough for a representative sample?

Thanks for your comments. We updated the manuscript by adding more recorded neurons (Figure 5A-F). The successfully recorded neurons were collected from three mice per group.

– I am confused why glucose seemingly inhibits MC4R neurons. If AgRP neurons inhibit MC4R, and glucose inhibits AgRP neurons, shouldn't they be activated?

According an in vitro ephys study the increase of extracellular glucose depolarized AgRP neurons that may mimic the hyperglycemic state in our experiment (6). Therefore, it is reasonable that the MC4R neurons in the DMH were inhibited after glucose treatment due to the activation of AgRP neurons.

5) Some of the histological data lack quantification.

We added quantification data and updated our manuscript.

6) Overall, the language throughout the manuscript is very operational, especially when describing the experiments. I would recommend conceptualizing the experiments, in addition to describing the tools (both within the text and in the figure legends) to improve flow and readability.

Per the suggestions, we have improved the writing in our manuscript.

References

Y. Y. Fang, T. Yamaguchi, S. C. Song, N. X. Tritsch, D. Lin, A Hypothalamic Midbrain Pathway Essential for Driving Maternal Behaviors. Neuron 98, 192-207 e110 (2018).G. Ugolini, H. G. Kuypers, A. Simmons, Retrograde transneuronal transfer of herpes simplex virus type 1 (HSV 1) from motoneurones. *Brain Res* 422, 242-256 (1987).C. L. Tan *et al.*, Warm-Sensitive Neurons that Control Body Temperature. *Cell* 167, 47-59.e15 (2016).J. Xu *et al.*, Genetic identification of leptin neural circuits in energy and glucose homeostases. *Nature* 556, 505-509 (2018).A. M. Aravanis *et al.*, An optical neural interface: in vivo control of rodent motor cortex with integrated fiberoptic and optogenetic technology. *J Neural Eng* 4, S143-156 (2007).M. Claret *et al.*, AMPK is essential for energy homeostasis regulation and glucose sensing by POMC and AgRP neurons. *J Clin Invest* 117, 2325-2336 (2007).

[Editors’ note: what follows is the authors’ response to the second round of review.]

The manuscript has been improved but there are some remaining issues that need to be addressed, as outlined below.Both reviewers felt strongly that there are several claims that are not substantiated in the manuscript and that all remaining comments need to be addressed in full prior to re-evaluation. The full comments are detailed below.Reviewer #2 (Recommendations for the authors):The authors have added new data to the manuscript, including some important controls, and the manuscript is much improved.Some issues remain, however:Figure 1 (or appropriate supplement)- should show nsCre reporter activity in nsCre mice that have not received NB124.

Thanks for your positive comments on our studies. We added histological data in Figure 1—figure supplement 1 showing no tdTomato+ neurons observed in nsCre+ mice that have not received NB124.

Supplemental Figure 1- is pSTAT3 normal in other areas (e.g., the DMH, VMH)?

We examined the pSTAT3 level in DMH and VMH (Figure 1—figure supplement 3). We found that there were no significant differences between control and Agrp-Lepr KO mice.

Figure 3- the biccuculine experiment still suffers from the unknown extent of diffusion. Perhaps staining for FOS might show the extent of diffusion (based on the absence of FOS).

Thanks for your suggestion. We performed Fos immunostaining in the DMH and neighboring brain regions such as PLH, VMH, and ARC after bicuculline infusion experiment (Figure 4—figure supplement 2). Even though we observed a mild increase of Fos expression in the PLH, there is no significant difference. Therefore, the bicuculline showed minimal effects on neighboring brain regions in our study.

Figure 4M-O- as noted previously, expressing Flp-dependent WGA-Cre in Npy neurons may also delete Gad1/2 in these neurons. It should also cause deletion in other regions that are targeted by Agrp neurons. Thus, this is not DMH-specific.

We appreciate these valuable comments. Because this experiment about ablating GABA in DMH caused confusion to both reviewers and is not closely relevant to the scope of the manuscript, we removed this experiment from our updated manuscript as to best ensure the coherence and clarity of the main scheme.

While the finding that deleting Gabra3 in the DMH blocks leptin-induced anorexia is useful, it would be better to show that this maneuver blocks the effects of deleting Lepr from Agrp neurons or some other Agrp neuron-specific manipulation.

We appreciate these valuable comments. We performed a new experiment with injection of *AAV9*-*fDIO-ChR2-EYFP* into the ARC, *AAV9-DIO-Cas9-mCitrine* and *AAV9-*

*Gabra3^sgRNA^-tdTomato* into the DMH in the *Npy^Flp^::Mc4^Cre^* mice, which may help examine whether the deletion of α3-GABAA receptor signaling from the DMH blocks the effects induced by activation of AgRP-DMH circuit (Figure 5—figure supplement 2). Our data showed that AgRP-DMH circuit regulates feeding and glucose tolerance depending on the α3-GABAA receptor signaling in the MC4R^DMH^ neurons.

Reviewer #3 (Recommendations for the authors):The authors addressed some of my previous concerns and have also provided several new verifications for techniques which improves confidence in the manuscript. Some concerns remain which are described below. Another major issue is that, unless I missed it, there is no methods section included in the revised manuscript, making it difficult to evaluate many of the new experiments.

Thanks for these positive comments on our studies. In the revised manuscript, we updated the Methods section including all necessary information.

Specific remaining issues:– While the authors revised the manuscript using a lower dose of bicuculline, significant effects remain compared to controls for food intake, feed efficiency, and glucose tolerance, and therefore the original concerns remain for Figure 1.

Thanks for your comments. We applied a lower dose of bicuculline that was ineffective on the control mice but blocked the effects of the Agrp-Lepr KO mice (Figure 1J-N).

In Figure 4, analyses from repeated measures ANOVA were not reported and so it is unclear if there are differences between control and biculculline groups, and given the lack of methods here the experimental paradigm is unclear (e.g., what is a chronic, 5-min injection via osmotic minipump? – Figure 4 legend)?

We updated methods and ANOVA analysis in Figure 4 legend.

– The fact that the HSV results yielded 2x the overlap as the CTB experiment raises concern about using it as an anterograde tracer, as previously noted by another reviewer. This should be discussed at the very least, or perhaps the HSV results should be omitted completely.

Thanks for your comments. We discussed in the updated manuscript that HSV could be used as a reliable retrograde tracing method. (Line 249-256)

– The authors' editorial attempts to address concerns about site specificity are not very compelling and this is important as ARC and DMH are relatively near each other (and DMH is close to the ventricle – where infused substances could diffuse). Data demonstrating that optogenetic stimulation only activates AgRP neurons that project to DMH, and that infused substances do not reach 4V or ARC, would be helpful, and if this is not possible a discussion of these limitations is warranted.

Thanks for your comments. As ARC and DMH are relatively near each other, we examined the Fos expression in the ARC after activation of AgRP fiber in the DMH (Figure 4—figure supplement 1). We did not observe significant changes of Fos expression in the ARC.

To check if infusion of bicuculline into DMH affected brain regions around 4V or ARC, we performed Fos immunostaining in the DMH and neighboring brain regions such as PLH, VMH, and ARC after bicuculline infusion experiment (Figure 4—figure supplement 2). We did not observe significant difference of Fos expression in these regions.

Therefore, the BIC infused into DMH showed efficient GABAA antagonism to block LepR-mediated function, while showing the minimal effects on neighboring brain regions.

– I am still confused by the recordings of MC4R neurons – where both foraging and hyperglycemia inhibit MC4R neurons. The authors cite one study suggesting that glucose activates AgRP neurons but do not discuss the many other studies that show the opposite. At the very least this should be discussed rather than the conclusion being that "these findings are consistent with MC4R-DMH neurons' mediation of food intake and blood glucose"

Thanks for your comments. We added discussion to the manuscript accordingly (Line 283-293).

– I am also still confused with how the DMH GABA KO studies fit into the narrative of the paper. The effect size here is quite large (and it is an interesting finding), but the leptin signaling is systemic and not specific to AgRP neurons.

We appreciate these valuable comments. Because this experiment about ablating GABA in DMH caused confusion to both reviewers and is not closely relevant to the scope of the manuscript, we removed this experiment from our updated manuscript as to best ensure the coherence and clarity of the main scheme.